# A multidisciplinary approach for investigating dietary and medicinal habits of the Medieval population of Santa Severa (7th-15th centuries, Rome, Italy)

Angelo Gismondi[1☯], Marica Baldoni[2☯], Micaela Gnes[2], Gabriele Scorrano[2¤], Alessia D'Agostino[1], Gabriele Di Marco[1], Giulietta Calabria[2], Michela Petrucci[2], Gundula Müldner[3], Matthew Von Tersch[4], Alessandra Nardi[5], Flavio Enei[6], Antonella Canini[1], Olga Rickards[2], Michelle Alexander[4], Cristina Martínez-Labarga[2]*

1 Laboratory of Botany, Department of Biology, University of Rome "Tor Vergata", Rome, Italy, 2 Centro di Antropologia Molecolare per lo Studio del DNA antico, Department of Biology, University of Rome "Tor Vergata", Rome, Italy, 3 Department of Archaeology, University of Reading, Reading, England, United Kingdom, 4 Department of Archaeology, BioArCh, University of York, York, England, United Kingdom, 5 Department of Mathematics, University of Rome "Tor Vergata", Rome, Italy, 6 Museo Civico di Santa Marinella "Museo del Mare e della Navigazione Antica", Castello di Santa Severa (Roma–Italia)

☯ These authors contributed equally to this work.
¤ Current address: Section for Evolutionary Genomics, GLOBE Institute, University of Copenhagen, Copenhagen K, Denmark.
* cristina.martinez@uniroma2.it

## Abstract

A multidisciplinary approach, combining stable isotope analysis from bone proteins and investigations on dental calculus using DNA analysis, light microscopy, and gas chromatography coupled with mass spectrometry, was applied to reconstruct dietary and medicinal habits of the individuals recovered in the cemetery of the Castle of Santa Severa (7th-15th centuries CE; Rome, Italy). Stable isotope analysis was performed on 120 humans, 41 faunal specimens and 8 charred seeds. Dental calculus analyses were carried out on 94 samples. Overall, isotope data indicated an omnivorous diet based on $C_3$-terrestrial protein, although some individuals possessed carbon values indicative of $C_4$ plant consumption. In terms of animal protein, the diet was probably based on cattle, sheep, pig and chicken products, as witnessed by the archaeozoological findings. Evidence from calculus suggested the consumption of $C_3$ cereals, Fabaceae, Fagaceae, milk and dairy products. Secondary metabolites of herbs and wine were also detected. The detection of marine fish ancient DNA, as well as of ω3 fatty acids in calculus, hypothesized the consumption of marine foodstuffs for this coastal population, despite the lack of a clear marine isotopic signal and the presence of polyunsaturated fatty acids in plant tissues. Moreover, the knowledge of ethnopharmacological tradition and the application of medicinal plants (e.g. *Punica granatum* L., *Ephedra* sp. L.) were also identified. The detection of artemisinin, known to have antimalarial properties, led to hypothesize the presence of malaria in the area. Altogether, the combined application of microscopy and biomolecular techniques provided an innovative reconstruction of Medieval lifeways in Central Italy.

**Data Availability Statement:** All relevant data are within the paper and its Supporting Information files.

**Funding:** 1. Funder name: Consiglio Nazionale delle Ricerche IPERION CH.it; Grant number: CIVITAS; Grant Recipient: Flavio Enei. 2. Funder name: European Union's Horizon 2020 research and innovation programme under the Marie Sklodowska-Curie; Grant number:751349; Grant Recipient: Gabriele Scorrano. 3. Funder name: Ministero dell'Istruzione, dell'Università e della Ricerca (MIUR); Grant number: MIUR Excellence Department Project; Grant Recipient: Department of Mathematics, Univ. of Rome "Tor Vergata"

**Competing interests:** The authors declare no conflict of interest.

## Introduction

Changes in dietary habits have played a paramount role in most human evolutionary milestones [1–2]. In particular, the significant socio-economic transformations that impacted on both Italy and Europe throughout the Medieval period, extensively affected food demand and dietary practice [3–8]. For example, the terrestrial-based subsistence typical of the earlier Middle Ages underwent important changes in the later Medieval period, during which an higher consumption of fish was generally favoured by the Christianity-related abstinence from meat in established periods of the year e.g. Fridays and during Lent [2–3, 7–18]. However, Medieval Italian food habits were known to be highly variable and tended to relate both to the social context and local environment of a particular population [7, 19–22].

The present investigation focuses on human remains found in the Medieval cemetery located near the Castle of Santa Severa, on the Latium coast, 50 Km North of Rome (42˚ 01′31″N 11˚56′54″E; Central Italy). The Medieval castle was built on the settlement of *Pyrgi*, an Etruscan harbour of the city of *Caere*, today known as *Cerveteri*. Archaeological excavation unearthed two cemetery areas, known as "*Casa del Nostromo*" and "*Piazza della Rocca*", in use from the 7th to the 15th century CE, as confirmed by radiocarbon analysis on four skeletal specimens performed at the Centre for Diagnostic and Dating (CEDAD) of the University of Salento [23]. The skeletal collection has undergone osteological analysis and consists of 455 individuals, comprising 188 non-adults and 267 adults, which, overall, showed a high degree of biomechanical stress related to daily activities [24]. Degenerative and infectious diseases were also documented in some individuals. Regarding dental health status, low frequency of abscesses, caries and *ante-mortem* tooth loss were documented. Based on the results of musculoskeletal stress markers obtained by the osteological analysis, the economy of this community was probably based on both farming and agricultural activities. The individuals were mainly buried wrapped in shrouds into simple earthen graves or in sarcophagi made up of re-used materials from pre-existing buildings. Grave goods were recovered only in some non-adult burials. The only exception is represented by the individual NS SU 321 which was buried in a sarcophagus where a cross and a stone cushion could be detected (see [24] for further details).

This research aims to provide an in-depth analysis of diet, lifeways and medicinal habits of the Medieval population of Santa Severa, through the integration of isotopic analysis of bone proteins and cut-edge archaeobotanical and molecular technologies on dental calculus, both highly informative in palaeodiet reconstruction [e.g. 25–28]. Carbon isotopes are useful to distinguish the consumption of $C_3$ *versus* $C_4$ plants and the contribution of marine foods in the diet [29–32]. The interpretation of the human data, is highly dependent on the local environmental context and in that perspective, the analysis of coeval faunal and plant remains, represents a fundamental requisite as the isotopic composition of foodstuffs varies both geographically and temporally [2, 17–18, 33].

Dental calculus, or mineralized oral plaque, is a dense mineral matrix that adheres to tooth surfaces [27, 34–35]. This deposit develops from inorganic salts, deriving from saliva, which trap organic molecules belonging to oral microbiota, foods, and non-dietary microremains (e.g. pollen grains), accidentally or intentionally inhaled/ingested during everyday activities (providing information on the surrounding environment) [27, 36]. Dental calculus tends to be well preserved in archaeological contexts [34] and is increasingly recognised as a valuable resource to investigate past human diet and phytotherapeutic practices [22, 37–43]. The analysis of dental calculus was mainly performed by light microscopy (henceforth LM); only rarely gas-chromatography-mass spectrometry (henceforth GC-MS) and occasionally genetic techniques have been combined [20, 22]. This research used this combination of techniques to

evaluate the role of plant species in phytotherapic practices and the extent of aquatic resources consumed by the coastal population of Santa Severa.

## Materials and methods

### Ethics statement

Specimen numbers of the skeletal remains analysed in the present research are reported in Tables 1–4 and in S1 Table. The archaeological collection is housed at Museo Civico di Santa Marinella "Museo del Mare e della Navigazione Antica", Castello di Santa Severa (Roma–Italia; responsible Dr. Flavio Enei). No permits were required for the described study, which complied with all relevant regulations. The responsible of the remains (Dr. Flavio Enei) is co-author of the present research.

### Protein extraction and isotope analysis

Adult individuals (n = 112), aged approximately 18 years and over, were selected for stable isotope analysis on the basis of their good preservation and the possibility to determine age and sex. Eight non-adults were also examined, for comparison between adult and non-adult diets. To better interpret the human isotope data, 41 adult faunal bones (6 *Bos* sp., 1 *Bubalus* sp., 1 *Equus* sp., 6 *Ovis* sp., 2 *Cervus* sp., 4 *Gallus* sp., 5 *Sus* sp., 1 *Sus scrofa*, 2 *Canis* sp., 1 *Felis* sp. and 12 fish, including 6 unidentified species—Pisces, 1 *Thunnus* sp., 1 *Galeorhinus* sp., 2 specimens of the Sparidae family, 1 *Labrus* sp. and 1 *Sparus* sp.; [44]) and 8 charred seeds (2 *Triticum* sp. L., 3 *Hordeum vulgare* L., 2 *Vicia* sp. L. and 1 *Vicia faba* L.; [45]) were also analysed. For each sample, fragments of bone (reported in Table 1) were cleaned by scraping, to remove potential external contaminants, and pulverized using pestle and mortar. Protein extraction was carried out as reported in Longin [46] with modifications [20, 47]. In brief, 0.5 g of powdered bones were demineralised in 0.6 M HCl for 2 days at 4°C on a horizontal mixer, replacing the HCl every 24 hours. Once all minerals were dissolved, samples were rinsed three times with bi-distilled water, until the pH became neutral. Then, sample gelatinization was performed in presence of 1 mM HCl, at 70°C, for 48 hours. The liquid fraction, containing gelatinized proteins, was frozen at -80°C for 4 hours and lyophilized. To evaluate the extraction efficiency, bone proteins of a modern bovine (with a known isotopic composition) were used as reference control for all protein extractions. For the seeds, no pre-treatment was used [48], they were homogenised using a pestle and mortar prior to analysis. In order to determine carbon ($\delta^{13}$C) and nitrogen ($\delta^{15}$N) isotope ratios for each specimen, 0.8–1.2 mg of proteins or 2 mg of homogenised grain were analysed, in duplicate, by continuous flow isotope ratio mass spectrometry (CF-IRMS) at the University of Reading (UK) or by EA/IRMS in a GSL analyser coupled to a 20–22 mass spectrometer (Sercon, Crewe, UK) at the University of York (UK). Samples were run alongside internal and international reference standards at both laboratories (in-house Fish gelatine: $\delta^{13}$C -15.5 ± 0.1‰, $\delta^{15}$N 14.3 ± 0.2‰; Cane sugar IA-R006: $\delta^{13}$C -11.8 ± 0.1‰; Caffeine IAEA 600: $\delta^{13}$C -27.8 ± 0.1‰, $\delta^{15}$N 0.8 ± 0.1‰; Ammonium Sulfate IAEA N2: $\delta^{15}$N 20.4 ± 0.2‰). The accepted analytical error was ≤0.2‰, for both carbon and nitrogen. To test reliability and exclude contamination events by exogenous carbon and nitrogen sources, the samples were assessed against established criteria [49–51]. Wilcoxon and Kruskal-Wallis tests were performed to statistically evaluate the obtained isotopic results. Average linkage cluster analysis was used to identify the presence of subgroups. Since variables with large variance tend to have a larger effect on the resulting clusters, variables were standardised before performing the analysis. All analyses were undertaken using SAS version 9.4 (SAS Institute, Cary NC).

**Table 1. Results of human and animal samples subjected to stable isotope analysis and quality indicators of protein extraction procedure were reported.** For each human sample, the examined bone and the biological information were provided. Faunal samples were classified at genus level (*Sus scrofa*: species; Pisces: not identified fish; Sparidae: family level). The specimens excluded from the present study were highlighted in red. For sex assessment, M indicates males, F indicates females, ND was used for not determined individuals (sex determination was impossible for the lack of diagnostic elements due to their poor state of preservation), while IND was applied to indeterminate individuals (non-adults whose sex was impossible to determine because of their sexual immaturity). GAs (generic adults) were the individuals whose age at death was impossible to determine for the lack of diagnostic elements due to their poor preservation state.

| | Human samples | | | | | | | | | |
|---|---|---|---|---|---|---|---|---|---|---|
| | Sample code | Examined bone | Sex | Age at death | $\delta^{13}$C, ‰ | $\delta^{15}$N, ‰ | %C | %N | C:N | % Protein yield |
| 1 | NS SU 14 Aa | Rib | F | 18–30 | -19.4 | 8.0 | 41.6 | 15.3 | 3.2 | 9 |
| 2 | NS SU 89 Aa | Rib | M | >50 | -19.2 | 9.5 | 42.4 | 15.7 | 3.2 | 13 |
| 3 | NS SU 89 Sc | Metatarsal | IND | 13–17 | -19.1 | 10.2 | 41.4 | 15.2 | 3.2 | 6 |
| 4 | NS SU 93 | Rib | M | 18–30 | -19.2 | 9.8 | 42.2 | 15.3 | 3.2 | 5 |
| 5 | NS SU 95 Aa | Rib | F | 18–30 | -19.2 | 10.3 | 53.3 | 19.1 | 3.3 | 7 |
| 6 | NS SU 99 | Rib | F | 18–30 | -19.0 | 8.9 | 42.4 | 15.1 | 3.3 | 5 |
| 7 | NS SU 104 Ab | Radius | M | 31–40 | -19.7 | 8.3 | 42.4 | 15.6 | 3.2 | 13 |
| 8 | NS SU 104 Sa | Rib | IND | 7–12 | -19.4 | 9.3 | 43.6 | 16.0 | 3.2 | 11 |
| 9 | NS SU 124 Aa | Rib | M | 31–40 | -16.8 | 10.4 | 42.1 | 15.4 | 3.2 | 6 |
| 10 | NS SU 126 | Rib | M | 31–40 | -19.2 | 8.5 | 52.0 | 18.4 | 3.3 | 5 |
| 11 | NS SU 135 | Metacarpal | M | >50 | -19.3 | 9.6 | 38.6 | 14.2 | 3.2 | 13 |
| 12 | NS SU 137 Aa | Humerus | M | 31–40 | -19.5 | 8.1 | 43.8 | 16.0 | 3.2 | 13 |
| 13 | NS SU 137 Ab | Humerus | M | 18–30 | -18.8 | 7.3 | 42.6 | 15.5 | 3.2 | 8 |
| 14 | NS SU 138 Ab | Rib | F | 41–50 | -19.2 | 9.6 | 41.9 | 15.4 | 3.2 | 7 |
| 15 | NS SU 139 Ab | Rib | M | 18–30 | -19.6 | 6.7 | 43.3 | 15.7 | 3.2 | 4 |
| 16 | NS SU 140 | Rib | F | 18–30 | -19.1 | 9.6 | 46.8 | 17.0 | 3.2 | 6 |
| 17 | NS SU 150 | Rib | M | 18–30 | -18.4 | 9.9 | 42.0 | 15.2 | 3.2 | 5 |
| 18 | NS SU 151 Aa | Rib | M | 18–30 | -18.8 | 8.0 | 42.3 | 15.3 | 3.2 | 7 |
| 19 | NS SU 155 | Rib | F | 31–40 | -19.0 | 10.5 | 45.1 | 16.6 | 3.2 | 17 |
| 20 | NS SU 157 Aa | Rib | F | 31–40 | -19.0 | 9.4 | 42.4 | 15.6 | 3.2 | 10 |
| 21 | NS SU 158 Aa | Rib | F | 18–30 | -18.9 | 7.7 | 43.1 | 15.6 | 3.2 | 6 |
| 22 | NS SU 163 Aa | Rib | F | 18–30 | -18.3 | 9.1 | 42.3 | 15.3 | 3.2 | 5 |
| 23 | NS SU 164 Aa | Rib | F | 18–30 | -18.0 | 9.1 | 42.4 | 15.4 | 3.2 | 7 |
| 24 | NS SU 165 | Rib | M | 18–30 | -19.3 | 6.8 | 43.4 | 15.7 | 3.2 | 2 |
| 25 | NS SU 171 | Rib | M | GA | -19.4 | 10.5 | 42.9 | 15.6 | 3.2 | 4 |
| 26 | NS SU 192 | Rib | M | 31–40 | -18.8 | 10.3 | 42.7 | 15.5 | 3.2 | 9 |
| 27 | NS SU 193 Aa | Rib | F | 31–40 | -19.0 | 9.4 | 43.8 | 15.8 | 3.2 | 8 |
| 28 | NS SU 196 Aa | Radius | F | 41–50 | -19.1 | 9.2 | 43.9 | 15.9 | 3.2 | 7 |
| 29 | NS SU 196 Ab | Radius | M | 18–30 | -19.8 | 10.3 | 42.1 | 15.2 | 3.2 | 6 |
| 30 | NS SU 200 Aa | Metatarsal | M | GA | -19.0 | 10.1 | 42.0 | 15.5 | 3.2 | 10 |
| 31 | NS SU 206 | Rib | M | 18–30 | -19.3 | 8.3 | 43.4 | 15.8 | 3.2 | 9 |
| 32 | NS SU 207 | Tibia | M | GA | -19.3 | 9.7 | 43.9 | 16.0 | 3.2 | 7 |
| 33 | NS SU 210 | Rib | M | 18–30 | -19.1 | 8.7 | 40.7 | 14.8 | 3.2 | 9 |
| 34 | NS SU 215 | Rib | F | 18–30 | -19.7 | 6.7 | 43.2 | 15.5 | 3.2 | 6 |
| 35 | NS SU 216 | Femur | M | 31–40 | -18.7 | 9.6 | 42.3 | 15.4 | 3.2 | 7 |
| 36 | NS SU 217 Aa | Rib | M | 18–30 | -19.0 | 9.5 | 44.6 | 16.3 | 3.2 | 11 |
| 37 | NS SU 224 | Rib | M | 41–50 | -19.2 | 10.8 | 42.0 | 15.3 | 3.2 | 6 |
| 38 | NS SU 225 Sa | Rib | IND | 13–17 | -19.1 | 9.3 | 41.6 | 15.2 | 3.2 | 6 |
| 39 | NS SU 227 | Rib | M | 31–40 | -18.9 | 9.7 | 41.4 | 15.2 | 3.2 | 5 |
| 40 | NS SU 230 | Rib | IND | 7–12 | -19.0 | 10.1 | 43.9 | 16.1 | 3.2 | 8 |
| 41 | NS SU 231 | Humerus | M | 31–40 | -18.9 | 10.4 | 43.1 | 15.4 | 3.3 | 2 |

*(Continued)*

**Table 1.** (Continued)

| 42 | NS SU 235 | Rib | M | 18–30 | -19.2 | 8.8 | 42.5 | 15.3 | 3.2 | 9 |
|----|-----------|-----|---|-------|-------|-----|------|------|-----|----|
| 43 | NS SU 237 Aa | Tibia | M | 41–50 | -19.1 | 8.5 | 43.4 | 15.8 | 3.2 | 11 |
| 44 | NS SU 238 Aa | Rib | F | 41–50 | -19.3 | 8.8 | 41.9 | 15.4 | 3.2 | 4 |
| 45 | NS SU 240 Aa | Humerus | M | 18–30 | -19.1 | 9.6 | 41.9 | 15.1 | 3.2 | 8 |
| 46 | NS SU 240 Ab | Humerus | F | GA | -18.9 | 8.9 | 43.2 | 15.6 | 3.2 | 3 |
| 47 | NS SU 241 | Rib | M | 31–40 | -18.9 | 9.0 | 42.3 | 15.5 | 3.2 | 10 |
| 48 | NS SU 248 | Rib | M | 18–30 | -19.3 | 10.1 | 39.1 | 14.3 | 3.2 | 8 |
| 49 | NS SU 285 Aa | Femur | F | 31–40 | -18.0 | 9.4 | 43.7 | 15.8 | 3.2 | 4 |
| 50 | NS SU 286 Aa | Rib | M | 41–50 | -19.1 | 8.9 | 43.5 | 16.0 | 3.2 | 12 |
| 51 | NS SU 287 Aa | Rib | M | 31–40 | -18.6 | 7.7 | 44.3 | 16.1 | 3.2 | 19 |
| 52 | NS SU 289 | Rib | M | 41–50 | -18.8 | 10.8 | 43.2 | 15.8 | 3.2 | 8 |
| 53 | NS SU 292 Aa | Metatarsal | M | >50 | -19.1 | 8.4 | 42.7 | 15.8 | 3.2 | 11 |
| 54 | NS SU 293–306 | Rib | M | 18–30 | -19.5 | 9.6 | 42.9 | 15.6 | 3.2 | 13 |
| 55 | NS SU 295 Aa | Femur | M | 41–50 | -19.7 | 7.6 | 42.8 | 15.6 | 3.2 | 12 |
| 56 | NS SU 295 Ab | Femur | M | GA | -18.9 | 7.7 | 40.3 | 14.9 | 3.2 | 11 |
| 57 | NS SU 299 Sa | Rib | IND | 1–6 | -19.3 | 10.1 | 44.7 | 16.4 | 3.2 | 14 |
| 58 | NS SU 304 Ac | Rib | M | 18–30 | -18.5 | 10.3 | 42.0 | 15.5 | 3.2 | 43 |
| 59 | NS SU 307 | Rib | F | 31–40 | -18.7 | 9.4 | 40.2 | 14.6 | 3.2 | 5 |
| 60 | NS SU 308 | Rib | F | 18–30 | -19.4 | 7.7 | 43.9 | 16.0 | 3.2 | 21 |
| 61 | NS SU 309 Aa | Rib | F | 41–50 | -18.3 | 10.5 | 43.0 | 15.8 | 3.2 | 9 |
| 62 | NS SU 310 Aa | Rib | F | 31–40 | -18.8 | 8.6 | 43.5 | 15.6 | 3.3 | 10 |
| 63 | NS SU 317 Aa | Rib | M | >50 | -18.8 | 10.1 | 43.7 | 15.9 | 3.2 | 14 |
| 64 | NS SU 318 Aa | Rib | M | 18–30 | -19.2 | 9.2 | 41.9 | 15.4 | 3.2 | 5 |
| 65 | NS SU 319 | Rib | F | 18–30 | -18.8 | 9.2 | 42.2 | 15.4 | 3.2 | 11 |
| 66 | NS SU 321 | Rib | M | >50 | -18.1 | 11.3 | 43.5 | 15.6 | 3.2 | 4 |
| 67 | NS SU 322 | Rib | F | 31–40 | -19.1 | 8.7 | 44.7 | 16.2 | 3.2 | 9 |
| 68 | NS SU 326 Aa | Rib | F | 31–40 | -19.3 | 8.5 | 54.7 | 19.6 | 3.2 | 8 |
| 69 | NS SU 327 Sa | Rib | F | 13–17 | -19.2 | 10.1 | 40.7 | 14.9 | 3.2 | 13 |
| 70 | NS SU 331 Aa | Rib | M | 31–40 | -18.8 | 9.8 | 41.3 | 15.1 | 3.2 | 14 |
| 71 | NS SU 336 Aa | Rib | M | 31–40 | -19.3 | 7.8 | 43.3 | 15.8 | 3.2 | 9 |
| 72 | NS SU 341 Aa | Rib | M | 18–30 | -19.2 | 9.6 | 41.5 | 15.3 | 3.2 | 13 |
| 73 | NS SU 346 Ab | Femur | F | 41–50 | -19.4 | 9.1 | 43.0 | 15.6 | 3.2 | 12 |
| 74 | NS SU 347 Ab | Rib | ND | 18–30 | -18.7 | 9.4 | 44.7 | 16.4 | 3.2 | 10 |
| 75 | NS SU 352 Aa | Rib | F | 18–30 | -19.3 | 9.7 | 43.7 | 15.9 | 3.2 | 7 |
| 76 | NS SU 354 Aa | Femur | M | GA | -18.9 | 9.6 | 39.9 | 14.5 | 3.2 | 8 |
| 77 | PR I SU 3 | Rib | F | 18–30 | -18.8 | 10.8 | 44.1 | 16.2 | 3.2 | 4 |
| 78 | PR I SU 4 Aa | Rib | F | 31–40 | -18.7 | 9.7 | 44.6 | 16.5 | 3.2 | 3 |
| 79 | PR I SU 5 | Rib | M | 31–40 | -18.8 | 8.1 | 42.2 | 15.4 | 3.2 | 5 |
| 80 | PR I SU 6 | Rib | F | 31–40 | -18.7 | 10.7 | 49.3 | 18.1 | 3.2 | 4 |
| 81 | PR I SU 7 | Rib | M | 18–30 | -19.0 | 9.1 | 56.3 | 20.8 | 3.2 | 4 |
| 82 | PR I SU 16 | Rib | F | 18–30 | -19.1 | 10.4 | 52.8 | 19.5 | 3.2 | 4 |
| 83 | PR I SU 17 | Metatarsal | M | GA | -19.3 | 10.1 | 43.4 | 15.8 | 3.2 | 7 |
| 84 | PR I SU 20 | Rib | M | 18–30 | -18.8 | 9.6 | 43.6 | 15.9 | 3.2 | 5 |
| 85 | PR I SU 37 | Rib | F | 18–30 | -18.8 | 9.7 | 46.9 | 17.2 | 3.2 | 5 |
| 86 | PR I SU 57 | Rib | F | 41–50 | -19.2 | 8.6 | 48.4 | 17.6 | 3.2 | 4 |
| 87 | PR I SU 58 | Rib | F | 31–40 | -18.5 | 7.8 | 46.5 | 17.1 | 3.2 | 3 |
| 88 | PR II SU 60 Aa | Rib | F | 41–50 | -19.1 | 9.1 | 45.8 | 16.7 | 3.2 | 9 |
| 89 | PR II SU 66 | Metatarsal | M | GA | -18.8 | 10.6 | 44.8 | 16.5 | 3.2 | 6 |

*(Continued)*

**Table 1.** (Continued)

| 90 | PR II SU 67 | Metatarsal | M | GA | -18.6 | 10.2 | 44.1 | 16.3 | 3.2 | 4 |
|----|-------------|------------|---|-----|-------|------|------|------|-----|---|
| 91 | PR I SU 106 | Metatarsal | M | GA | -19.2 | 7.0 | 44.6 | 16.2 | 3.2 | 5 |
| 92 | PR II SU 112 | Skull | M | 41–50 | -18.6 | 11.4 | 43.9 | 15.9 | 3.2 | 4 |
| 93 | PR I SU 117 | Metatarsal | F | GA | -19.7 | 9.5 | 43.5 | 15.9 | 3.2 | 4 |
| 94 | PR II SU 120 | Rib | ND | 18–30 | -18.7 | 7.2 | 42.1 | 15.3 | 3.2 | 5 |
| 95 | PR I SU 122 | Metatarsal | M | GA | -17.9 | 7.8 | 44.5 | 16.4 | 3.2 | 4 |
| 96 | PR I SU 123 | Femur | IND | 13–17 | -19.3 | 10.8 | 42.8 | 15.5 | 3.2 | 7 |
| 97 | PR I SU 124 Aa | Fibula | M | GA | -19.5 | 8.7 | 41.2 | 14.9 | 3.2 | 6 |
| 98 | PR I SU 124 Ab | Fibula | ND | GA | -19.2 | 10.0 | 42.7 | 15.6 | 3.2 | 12 |
| 99 | PR I SU 125 | Metatarsal | M | GA | -19.0 | 10.5 | 44.8 | 16.4 | 3.2 | 6 |
| 100 | PR I SU 126 | Metatarsal | F | GA | -19.3 | 9.3 | 44.5 | 16.4 | 3.2 | 4 |
| 101 | PR I SU 128 Aa | Metatarsal | M | GA | -19.0 | 10.2 | 43.5 | 15.9 | 3.2 | 3 |
| 102 | PR I SU 129 Aa | Fibula | F | GA | -18.5 | 9.0 | 43.8 | 16.2 | 3.2 | 7 |
| 103 | PR I SU 130 | Fibula | M | GA | -19.4 | 10.0 | 44.7 | 16.3 | 3.2 | 7 |
| 104 | PR II SU 133 Aa | Metatarsal | F | GA | -18.9 | 10.7 | 44.8 | 16.4 | 3.2 | 5 |
| 105 | PR II SU 135 | Metatarsal | IND | 13–17 | -18.6 | 7.7 | 41.6 | 15.2 | 3.2 | 7 |
| 106 | PR II SU 141 | Metatarsal | M | GA | -18.9 | 10.6 | 44.5 | 16.5 | 3.1 | 7 |
| 107 | PR II SU 144 Aa | Tibia | F | GA | -17.7 | 8.1 | 42.9 | 15.7 | 3.2 | 9 |
| 108 | PR II SU 144 Ab | Tibia | M | GA | -19.2 | 10.2 | 42.3 | 15.4 | 3.2 | 7 |
| 109 | PR II SU 147 | Metatarsal | M | GA | -18.9 | 8.5 | 43.6 | 16.0 | 3.2 | 7 |
| 110 | PR I SU 148 Aa | Tibia | F | GA | -19.0 | 7.2 | 42.6 | 15.8 | 3.2 | 16 |
| 111 | PR I SU 151 | Metatarsal | M | GA | -17.8 | 8.0 | 43.6 | 16.0 | 3.2 | 7 |
| 112 | PR I SU 152 | Metatarsal | F | GA | -18.8 | 10.7 | 43 | 15.8 | 3.2 | 4 |
| 113 | PR II SU 173 | Metatarsal | ND | 18–30 | -19.3 | 9.3 | 41 | 15.0 | 3.2 | 6 |
| 114 | PR III SU 236 | Rib | M | 31–40 | -19.3 | 9.0 | 46 | 16.7 | 3.2 | 4 |
| 115 | PR III SU 247 | Rib | M | 41–50 | -18.4 | 10.8 | 45.7 | 16.5 | 3.2 | 3 |
| 116 | PR III SU 257 | Rib | M | 18–30 | -19.1 | 9.3 | 42.5 | 15.3 | 3.2 | 4 |
| 117 | PR II SU 262 Aa | Tibia | M | GA | -17.5 | 9.9 | 43.4 | 15.9 | 3.2 | 10 |
| 118 | PR II SU 267 Aa | Rib | F | GA | -19.1 | 10.3 | 41.6 | 15.2 | 3.2 | 14 |
| 119 | PR II SU 267 Sb | Femur | F | 18–30 | -19.2 | 8.8 | 43.7 | 16.0 | 3.2 | 10 |
| 120 | PR IV SU 298 | Tibia | M | GA | -18.5 | 7.1 | 41.6 | 15.2 | 3.2 | 14 |

**Faunal samples**

| | Sample code | Classification | $\delta^{13}$C, ‰ | $\delta^{15}$N, ‰ | %C | %N | C:N | % Protein yield |
|----|-------------|----------------|------------------|------------------|------|------|-----|-----------------|
| 1 | SS-BOS 1 | *Bos* sp. | -19.1 | 6.2 | 42.9 | 15.8 | 3.2 | 10 |
| 2 | SS-BOS 2 | *Bos* sp. | -19.9 | 7.7 | 41.7 | 15.2 | 3.2 | 15 |
| 3 | SS-BOS3 | *Bos* sp. | -20.2 | 7.7 | 50.7 | 18.4 | 3.2 | 8 |
| 4 | SS-BOS 4 | *Bos* sp. | -18.8 | 6.9 | 41.4 | 15.0 | 3.2 | 10 |
| 5 | SS-BOS 5 | *Bos* sp. | -20.9 | 5.1 | 44.3 | 16.3 | 3.2 | 4 |
| 6 | NS SU 301 B | *Bos* sp. | -21.2 | 6.9 | 40.4 | 14.9 | 3.2 | 15 |
| 7 | SS-BUFF | *Bubalus* sp. | -18.4 | 8.6 | 42.6 | 15.3 | 3.2 | 7 |
| 8 | SS-HORSE | *Equus* sp. | -20.2 | 5.6 | 53.0 | 19.1 | 3.2 | 7 |
| 9 | SS-OVIS 1 | *Ovis* sp. | -21.2 | 6.4 | 42.6 | 15.6 | 3.2 | 14 |
| 10 | SS-OVIS 2 | *Ovis* sp. | -20.1 | 7.1 | 42.0 | 15.1 | 3.3 | 11 |
| 11 | SS.OVIS 3 | *Ovis* sp. | -21.6 | 4.9 | 42.0 | 15.1 | 3.2 | 11 |
| 12 | SS-OVIS 4 | *Ovis* sp. | -20.4 | 4.8 | 45.1 | 16.4 | 3.2 | 9 |
| 13 | SS-OVIS 5 | *Ovis* sp. | -20.6 | 6.3 | 47.1 | 16.9 | 3.3 | 7 |
| 14 | NS US 72 | *Ovis* sp. | -21.2 | 5.2 | 33.5 | 12.5 | 3.1 | 38 |
| 15 | SS-CERVUS 1 | *Cervus* sp. | -21.2 | 4.4 | 42.0 | 15.3 | 3.2 | 18 |

*(Continued)*

**Table 1.** (Continued)

| 16 | **SS-CERVUS 2** | *Cervus* sp. | -20.8 | 4.3 | 42.8 | 15.6 | 3.2 | 9 |
|---|---|---|---|---|---|---|---|---|
| 17 | **SS-GALLUS 1** | *Gallus* sp. | -19.6 | 10.6 | 50.7 | 18.4 | 3.2 | 4 |
| 18 | **SS-GALLUS 2** | *Gallus* sp. | -19.7 | 9.1 | 69.8 | 25.1 | 3.2 | 11 |
| 19 | **SS-GALLUS 3** | *Gallus* sp. | -19.5 | 7.8 | 47.1 | 17.1 | 3.2 | 9 |
| 20 | **SS-GALLUS 4** | *Gallus* sp. | -19.6 | 8.7 | 47.4 | 17.3 | 3.2 | 7 |
| 21 | **SS-SUS 1** | *Sus* sp. | -20.5 | 5.2 | 41.9 | 15.2 | 3.2 | 15 |
| 22 | **SS-SUS 2** | *Sus* sp. | -19.9 | 8.1 | 41.8 | 15.3 | 3.2 | 16 |
| 23 | **SS-SUS 3** | *Sus* sp. | -20.8 | 6.7 | 48.9 | 17.7 | 3.2 | 10 |
| 24 | **SS-SUS 4** | *Sus* sp. | -19.9 | 6.1 | 44.4 | 16.2 | 3.2 | 8 |
| 25 | **SS-SUS 5** | *Sus* sp. | -20.9 | 6.0 | 61.6 | 22.8 | 3.2 | 10 |
| 26 | **SS-WBOAR** | *Sus scrofa* | -19.8 | 8.2 | 43.0 | 15.6 | 3.2 | 9 |
| 27 | **SS-DOG 1** | *Canis* sp. | -19 | 9.1 | 41.5 | 14.9 | 3.3 | 3 |
| 28 | **SS-DOG 2** | *Canis* sp. | -17.8 | 9.9 | 44.5 | 16.3 | 3.2 | 8 |
| 29 | **SS-CAT** | *Felis* sp. | -19.2 | 9.5 | 47.3 | 17.4 | 3.2 | 13 |
| 30 | **TVG-S-FISH 3** | Pisces | -7.9 | 9.8 | 49.7 | 18.6 | 3.1 | 10 |
| 31 | **TVG-S-FISH 5** | Pisces | -9.8 | 11.4 | 47.4 | 17.8 | 3.1 | 13 |
| 32 | **TVG-S-FISH 7** | Pisces | -9.6 | 11.5 | 50.2 | 18.7 | 3.1 | 10 |
| 33 | **TVG-SS-S FISH 8** | Pisces | -9.8 | 12.2 | 42.1 | 15.1 | 3.2 | 3 |
| 34 | **TVG-SS-S FISH 9** | Pisces | -9.8 | 11.2 | 44.5 | 16.4 | 3.2 | 8 |
| 35 | **TVG-S-TUNA** | *Thunnus* sp. | -13.8 | 13.3 | 55.3 | 17.9 | 3.6 | 8 |
| 36 | **SSF1** | Pisces | -19.8 | 7.6 | 42.9 | 15.6 | 3.2 | 14 |
| 37 | **SSF2** | *Galeorhinus* sp. | -12.0 | 13.9 | 41.9 | 14.8 | 3.3 | 11 |
| 38 | **SSF3** | Sparidae | -8.8 | 10.7 | 43.9 | 16.5 | 3.2 | 8 |
| 39 | **SSF4** | Sparidae | -9.3 | 10.4 | 42.9 | 16.1 | 3.1 | 9 |
| 40 | **SSF5** | *Labrus* sp. | -10.3 | 10.1 | 43.1 | 15.3 | 3.2 | 5 |
| 41 | **SSF6** | *Sparus* sp. | -7.3 | 9.7 | 41.7 | 15.3 | 3.2 | 2 |

## Molecular and archaeobotanical analyses on dental calculus

**Sample collection and decontamination.** Dental calculus flakes were collected from 94 individuals with an autoclaved dental pick, recovering the maximum amount of sample available per individual. Biological details of the individuals, calculus location and weight, and the analysis performed per each sample are reported in S1 Table. Dental calculus was frequently collected from multiple different teeth per individual, pooled and subsequently divided into sub-samples to be investigated. The samples were placed in sterile microcentrifuge tubes and preserved at the Department of Biology of the University of Rome 'Tor Vergata' (Italy). According to Crowther et al. [52], Gismondi et al. [53] and Soto et al. [54], an intensive regime of cleaning, using 5% sodium hypochlorite (NaClO), 5% sodium hydroxide (NaOH) and boiling sterilized water, was applied on surfaces, instruments and floor of all workspaces. Horizontal slide traps placed in several areas of the laboratories were monitored for the evaluation of the contaminants (S2 Table). Decontamination and sterilization protocols were conducted on the mineralised plaque, to eliminate soil particles still adhering to the external surface of the samples by a sterile acupuncture needle under a stereomicroscope (Leica ZOOM 2000, at a magnification of 30X). Each sample was treated with UV light for 10 min and immersed in 1 mL of 2% NaOH for 15 min. The pellet was then washed twice in sterile bidistilled water (40˚C), rinsed in 50 μL of 100% ethanol, and left to evaporate, at 37˚C, under a sterile vertical laminar flow hood (Heraeus HERAsafe HS12 Type). Before the application of the cleaning procedure, ten random dental calculus samples were selected and washed by sterile water; the

**Table 2. Stable isotope data of seed samples and relative weights and quality indicators.**

| | Sample code | Genus/Species | Weight (mg) | δ¹³C, ‰ | δ¹⁵N, ‰ | %C | %N | C:N |
|---|---|---|---|---|---|---|---|---|
| 1 | NS SU350 | *Hordeum vulgare* | 17.4 | -23.3 | 6.7 | 49.9 | 2.5 | 23.4 |
| 2 | NS SU 215 | *Hordeum vulgare* | 18.5 | -23.6 | 4.8 | 44.3 | 2.0 | 26.0 |
| 3 | NS SU 254 | *Hordeum vulgare* | 20.4 | -23.5 | 5.4 | 46.9 | 1.4 | 38.6 |
| 4 | NS SU 350 | *Triticum* sp. | 26.2 | -22.2 | 4.1 | 45.1 | 2.4 | 22.1 |
| 5 | NS SU 215 | *Triticum* sp. | 20.9 | -22.7 | 4.5 | 42.5 | 2.6 | 18.7 |
| 6 | NS SU 350 | *Vicia faba* | 66.0 | -22.2 | 2.8 | 43.2 | 4.9 | 10.3 |
| 7 | NS SU 299 | *Vicia* sp. | 18.0 | -21.0 | 2.5 | 42.0 | 4.9 | 10.0 |
| 8 | NS SU 350 | *Vicia* sp. | 12.3 | -22.4 | 1.9 | 43.4 | 5.0 | 10.1 |

latter was examined by optic microscopy. The results of these observations are reported in S3 Table. To confirm the efficacy of the proposed decontamination method, the same samples, after sterilization, were washed again by sterile water and subjected to microscopy analysis. No microremains were detected in this second washing water.

**DNA extraction, amplification and sequencing.** Dental calculus was powdered using a pestle and mortar. To minimize contamination, the aDNA analysis was carried out in the aDNA Laboratory in the Departmental Center of Molecular Anthropology for Ancient DNA Studies, University of Rome, "Tor Vergata", in Villa Mondragone, Monte Porzio Catone, Rome (http://www.bio.uniroma2.it/biologia/laboratori/lab-antropologia/DNAantico/DNA_antico/Facilities.htm) [55]. At least two independent DNA extraction were performed on 52 samples, following the protocol modified from Warinner and collaborators [56], as a minimum of 50 mg of tartar was necessary for this analysis. All criteria and precautions for ancient DNA (aDNA) study were applied [55, 57–59]. Negative control extraction and amplifications were performed. To each sample, 600 μL of extraction buffer (100 mM Tris/HCl pH 8, 100 mM NaCl, 10 mM EDTA pH 8 and 2% SDS) and 50 μl of proteinase K (20 mg/mL) were added. Samples were kept at a temperature of 56˚C for 6 hours and 10 μL of proteinase K (20 mg/ml) were added every two hours. After that, they were kept at 37˚C overnight. After centrifugation at 13000 rpm for 5 minutes, 500 μL of phenol/chloroform/isoamyl alcohol (25:24:1) were added to the liquid phase, followed by another centrifugation. The supernatant was then purified using QIAquick PCR purification kit following the manufacturer's procedure. Finally, DNA was eluted in 50 μL of EB Buffer. For each investigated region (*Ovis aries*, *Gallus gallus*, *Bos taurus* and *Sus scrofa domesticus*) at least two independent amplifications by PCR (Polymerase Chain Reaction) were carried out using species-specific primers and barcode primers for marine fish [60]. For animals, amplified sequences represented portions of COXI or 12S/16S rRNA genes of mitochondrial DNA (mtDNA) [61]. Before amplifying aDNA, a "Spike PCR" was performed to verify the absence of polymerase inhibitors. PCR mix was prepared in the following way: 3 μL of template DNA; 2.5 μL of PCR Buffer (10X); 2.5 μL of MgCl₂ (25 mM); 2.5 μL of dNTPs (10 mM); 1 μL of forward Primer (1 μM); 1 μL of reverse Primer (1 μM); 0.25 μL of BSA (10 mg/mL); 0.2 μL of Taq polymerase (5 U/μL); 12.05 μL of ddH₂O; for a total volume of 25 μL. The primers used for the "Spike PCR" and aDNA amplification are specified in S4 Table. PCR products were evaluated on 1.5% agarose gel stained with GelStar and, then, positive amplifications were purified using the ExoSAP method. After labelling with fluorescent nucleotides (Big Dye Terminator chemistry) and purification by a standard ethanol precipitation technique, samples were finally sequenced. The obtained sequences were compared with reference genes registered in GenBank, using BLAST software (https://blast.ncbi.nlm.nih.gov/Blast.cgi?PROGRAM=blastn&PAGE_TYPE=BlastSearch&LINK_LOC=blasthome).

**Table 3. Results of the DNA barcode analysis on dental calculus; the symbol "+" indicated positive PCR amplification.**

| | Sample code | Bovine | Pig | Ovine | Chicken | Fishes |
|---|---|---|---|---|---|---|
| 1 | NS SU 27 Ac | | | | | + |
| 2 | NS SU 78 Aa | | | + | | + |
| 3 | NS SU 82 | + | | + | | |
| 4 | NS SU 95 Aa | + | | | | + |
| 5 | NS SU 99 | + | | | | + |
| 6 | NS SU 104 Ab | | | | | + |
| 7 | NS SU 115 | | + | + | | + |
| 8 | NS SU 124 Aa | + | | | | + |
| 9 | NS SU 137 Ab | | + | + | | + |
| 10 | NS SU 140 | | | | | |
| 11 | NS SU 150 | | | | | |
| 12 | NS SU 151 Aa | | | | | + |
| 13 | NS SU 154 Aa | + | | + | | + |
| 14 | NS SU 155 | + | + | + | + | + |
| 15 | NS SU 157 Aa | | | | | + |
| 16 | NS SU 158 Aa | | | | | + |
| 17 | NS SU 164 Ab | | | | | |
| 18 | NS SU 165 | | | | | + |
| 19 | NS SU 168 | | | | | + |
| 20 | NS SU 177 Ab | | | | | + |
| 21 | NS SU 179 Aa | + | | | | + |
| 22 | NS SU 190 Ad | | | + | | + |
| 23 | NS SU 215 | | | | | + |
| 24 | NS SU 217 Aa | | + | + | + | + |
| 25 | NS SU 221 Aa | | | + | | + |
| 26 | NS SU 231 | + | + | + | | + |
| 27 | NS SU 235 | + | | + | | + |
| 28 | NS SU 240 Aa | | | + | | + |
| 29 | NS SU 241 | + | + | + | + | + |
| 30 | NS SU 283 Aa | | | + | | |
| 31 | NS SU 283 Ab | | | | | + |
| 32 | NS SU 286 Aa | + | | | | + |
| 33 | NS SU 287 Aa | + | + | + | + | + |
| 34 | NS SU 287 Ac | | | | | + |
| 35 | NS SU 290 | + | + | + | + | + |
| 36 | NS SU 292 Aa | | | | | + |
| 37 | NS SU 293–306 | + | | + | | + |
| 38 | NS SU 302 Aa | | | | | + |
| 39 | NS SU 307 | + | + | + | | + |
| 40 | NS SU 309 Aa | + | + | + | | + |
| 41 | NS SU 310 Aa | | | + | | + |
| 42 | NS SU 311 Aa | | | + | | + |
| 43 | NS SU 318 Aa | + | | + | | + |
| 44 | NS SU 319 | | + | + | | + |
| 45 | NS SU 321 | + | + | + | + | + |
| 46 | NS SU 322 | | | + | | + |
| 47 | NS SU 326 Aa | | | + | | |

*(Continued)*

**Table 3.** (Continued)

| Sample code | | Bovine | Pig | Ovine | Chicken | Fishes |
|---|---|---|---|---|---|---|
| 48 | NS SU 327 Sa | | | + | | + |
| 49 | NS SU 328 Ab | | | | | + |
| 50 | NS SU 341 Aa | | | + | | + |
| 51 | NS SU 347 Ab | | + | + | | + |
| 52 | NS SU 356 | | | | | + |
| Total | | 18 | 13 | 29 | 6 | 46 |
| Percentage (%) | | 34.6 | 25 | 55.8 | 11.5 | 88.5 |

In order to detect the correct activity of the different reagents and to check their contamination amplification were also performed on positive and negative controls.

**Light microscopy analysis (LM).** In the laboratories of Botany, specifically reserved to the analysis of ancient biomolecules, a decalcification procedure was employed to extrapolate micro-remains from dental calculus. Twenty mg per each of the 94 specimen were degraded in 0.5 mL of 0.2 M hydrochloric acid (HCl), for 8 hours, and, after 3 washings with bidistilled water, the pellet was resuspended in 100 μL of bidistilled water and glycerol (1:1), under a sterile hood, and placed on glass slides to be analysed at OM (Axio Observer 7, ZEISS). The whole sample volume was analysed and each micro-debris was measured and photographed using ZEN imaging software. All microfossils were recognised on the basis of their morphology and distinctive features by direct comparison with a modern experimental collection of starches [62] and literature data [63–64].

**Gas-Chromatography Mass-Spectrometry analysis (GC-MS).** GC-MS analysis was carried out for the individuals presenting a minimum of 10 mg of tartar (68 individuals). This qualitative approach had already been successfully applied by the authors, both on ancient dental calculus and Roman frescoes [22, 65]. Briefly, once dissolved in 0.5 mL of 3% HCl, samples were incubated with 0.5 mL di hexane and left in agitation for two hours. Then, the supernatant fraction was recovered and dried out by a speed-vac system (Eppendorf AG 22331 Hamburg, Concentration Plus). After resuspension in 60 μL of hexane and derivatization by 40 μL of Methyl-8-Reagent (Thermo Scientific), 2 μL of extract was injected in a GC-MS QP2010 system (Shimadzu, Japan), equipped with a DB-5 capillary column (Phenomenex; length 30 m × diameter 0.25 mm × thickness 0.25μm), at the temperature of 280˚C (splitless modality), in triplicate per each dental calculus. The carrier gas was helium (constant flow of 1 mL/min). Column temperature was initially set at 60˚C for 5 min (initial oven temperature) and, then, increased at a rate 6˚C/min up to 150˚C for 5 min, then up to 250˚C for 5 min, and to the final temperature of 330˚C for 25 min. An electron impact of 70 eV (scanning from 100 to 700 m/z) was used for the ionization (ion source temperature 230˚C, interface temperature 320˚C, solvent cut time 6 min). Each chemical compound was identified by comparison of its mass spectrum with those registered in the NIST Library 14 loaded in the detection software (similarity values higher than 85%). Specific plants and food categories, ingested at least once in lifetime, were supposed combining the detected analytes with data from the literature and scientific food databases[66–67]. No relevant differences among replicates were detected.

## Results

### Stable isotope analysis

Proteins were successfully extracted from all 120 human and 41 animal bones. All stable isotope results, including the relative quality control indicators, are reported in Table 1.

**Table 4. Plant microremains found in dental calculus samples by LM.** The amount of starch granules detected by direct microscopy observation in each sample were reported and classified per morphotype. Moreover, the total of the other microdebris identified in ancient dental calculus were indicated as follows: Oleaceae pollen grain (O), Chenopodiaceae pollen grain (C), Urticaceae pollen grain (U), Poaceae phytoliths (P), Calcium oxalate crystals (Z), Asteraceae inflorescence (A) and not determined pollen grains (ND).

| Sample name | Morphotype I | Morphotype II | Morphotype III | Morphotype IV | Morphotype V | Morphotype VI | Morphotype VII | Morphotype VIII | Not determined | TOTAL starch granules per individual | Others microdebris |
|---|---|---|---|---|---|---|---|---|---|---|---|
| NS SU 14 Aa | | | | | | | | | 1 | 1 | 1O |
| NS SU 15 Ad | | | | | | | | | 1 | 1 | 1C |
| NS SU 27 Ac | | 1 | | | | | | | | 1 | |
| NS SU 69 Ab | | | 43 | | | | | | | 43 | |
| NSSU 78 Aa | | | | | | | | | | 0 | |
| NS SU 78 Ag | | 2 | | | | | | 1 | | 3 | 1U |
| NS SU 78 Sb | | | 2 | | | | | | | 2 | |
| NS SU 78 Sc | | 1 | | | | | | | | 1 | |
| NS SU 85 Sc | | | 1 | | | | | | 1 | 2 | |
| NS SU 95 Aa | | | | 30 | | 3 | | | | 33 | |
| NS SU 99 | | | | | | | | | 1 | 1 | |
| NS SU 104 Ab | 1 | | | 1 | | | | | | 2 | 1P |
| NS SU 115 | | | 2 | | | | | 2 | 2 | 6 | 1P, 1A |
| NS SU 118 A | | | | | | 1 | 1 | 1 | 2 | 5 | 4P |
| NS SU 124 Aa | | | | | | | | 1 | | 1 | 2P |
| NS SU 129 Aa | | | | | | 1 | | | | 1 | |
| NS SU 129 Ac | | | | | | | | 1 | | 1 | |
| NS SU 130 Aa | | | | | | | | 1 | | 1 | 1P |
| NS SU 137 Ab | | | 1 | | | | | 1 | | 2 | |
| NS SU 138 Ab | | | 1 | | | | | | 1 | 2 | |
| NS SU 140 | | | 1 | | | | | | 1 | 2 | |
| NS SU 150 | 1 | | | | | | | | 1 | 2 | |
| NS SU 151 Aa | | | | 2 | | | | 1 | | 3 | |
| NS SU 154 Aa | | | | 1 | | | | | 2 | 3 | |
| NS SU 154 Ab | | | 3 | | | 3 | | | 2 | 8 | |
| NS SU 155 | | 4 | 2 | | | | | | 4 | 10 | |
| NS SU 157 Aa | | | 1 | | | | | | | 1 | |
| NS SU 158 Aa | | | 3 | | | | | | | 3 | |
| NS SU 163 Ab | | | | | | | | 1 | | 1 | |
| NS SU 164 Ab | | | | | | | | | | 0 | |
| NS SU 165 | | | 1 | | 1 | | | | | 2 | |
| NS SU 168 | | | | | | | | | | 0 | |
| NS SU 169 | | | | | | | | | | 0 | |
| NS SU 177 Aa | | | | | | | | | 2 | 2 | |
| NS SU 177 Ab | | | | | | | | | | 0 | |
| NS SU 179 Aa | 1 | 1 | | | | | | | 1 | 3 | |
| NS SU 190 Aa | | | 3 | 4 | | | | | 1 | 8 | |

*(Continued)*

Table 4. (Continued)

| Sample name | Morphotype I | Morphotype II | Morphotype III | Morphotype IV | Morphotype V | Morphotype VI | Morphotype VII | Morphotype VIII | Not determined | TOTAL starch granules per individual | Others microdebris |
|---|---|---|---|---|---|---|---|---|---|---|---|
| NS SU 190 Ad | | | | | | 25 | | | 3 | 28 | |
| NS SU 192 | | | | | | | | | | 0 | |
| NS SU 193 Aa | | | | | | | | | | 0 | |
| NS SU 193 Ab | | | 2 | | | | | | | 2 | |
| NS SU 196 Aa | 1 | | 4 | 2 | | | | | | 7 | |
| NS SU 202 | 2 | | | | | | | | | 2 | |
| NS SU 205 Ac | | | | | | | | | 1 | 1 | |
| NS SU 210 | | | | | | | | | | 0 | |
| NS SU 215 | | | 2 | | | | 1 | | 1 | 4 | 2O |
| NS SU 217 Aa | | | | 6 | | | | | | 6 | 7P, 2ND |
| NS SU 219 Aa | | | | | | | | | | 0 | |
| NS SU 221 Aa | | | | | | | | | 2 | 2 | |
| NS SU 231 | | | | | | | | | 1 | 1 | 1P, 2Z |
| NS SU 235 | 2 | | 1 | | 1 | 2 | | | 1 | 7 | 1P |
| NS SU 237 Aa | | | | | | | | | | 0 | |
| NS SU 240 Aa | 1 | | | | | | | | 1 | 2 | |
| NS SU 241 | 1 | | 8 | | 1 | | | | | 10 | 1ND |
| NS SU 283 Aa | | | | | | | | | | 0 | |
| NS SU 283 Ab | | | | | | | | | | 0 | |
| NS SU 284 Aa | 1 | 1 | 1 | | | | | | | 3 | |
| NS SU 286 Aa | | | 1 | | | | | | 3 | 4 | |
| NS SU 287 Aa | | | | | | | | | | 0 | |
| NS SU 287 Ab | | | | | | | | | | 0 | |
| NS SU 287 Ac | | | | 1 | | | | | | 1 | |
| NS SU 289 | | | | | | | | | | 0 | |
| NS SU 290 | | | 1 | | | | | | 2 | 3 | 1A |
| NS SU 292 Aa | | 3 | | | | | | | | 0 | |
| NS SU 293–306 | | | | | | | | | 2 | 5 | |
| NS SU 299 Sa | | | | | | | | | | 0 | |
| NS SU 301 Ab | | | | | | | | | 1 | 1 | |
| NS SU 302 Aa | | | | | | 1 | | | | 1 | |
| NS SU 302 Ab | | | 1 | | | | | | | 1 | |
| NS SU 304 Aa | | 4 | | | | | | | | 0 | |
| NS SU 307 | | | | | | | | | 1 | 5 | |
| NS SU 308 | | | | | | | | | 1 | 1 | |
| NS SU 309 Aa | 2 | | 7 | 3 | | 1 | | | 7 | 20 | 1P |
| NS SU 310 Aa | | | | | | | | 1 | 1 | 2 | |
| NS SU 311 Aa | | | | | | | | 1 | | 1 | 1ND |

(Continued)

**Table 4.** (Continued)

| Sample name | Morphotype I | Morphotype II | Morphotype III | Morphotype IV | Morphotype V | Morphotype VI | Morphotype VII | Morphotype VIII | Not determined | TOTAL starch granules per individual | Others microdebris |
|---|---|---|---|---|---|---|---|---|---|---|---|
| NS SU 316 Aa | | | | | | | | | | 0 | |
| NS SU 317 Aa | | | | | | | | | | 0 | |
| NS SU 318 Aa | | 2 | | | | | | | 1 | 3 | |
| NS SU 319 | 1 | | | | | | | | 1 | 2 | 1Z |
| NS SU 320 | | | | 2 | | | | | | 2 | |
| NS SU 321 | 6 | 8 | | | | 3 | | | 4 | 21 | |
| NS SU 322 | | 1 | 7 | 2 | | | | | 2 | 12 | |
| NS SU 325 | | | | | | | | | | 0 | |
| NS SU 326 Aa | 1 | 1 | 1 | | | | | | 1 | 4 | |
| NS SU 327 Sa | 1 | | | | | | | | | 1 | |
| NS SU 331 Aa | | | 1 | | | | | | | 1 | |
| NS SU 333 | 1 | | 1 | | | | | | | 2 | |
| NS SU 336 Aa | | | | | | | | | | 0 | |
| NS SU 341 Aa | | | | | | | | | | 0 | |
| NS SU 344 Aa | | | | | | | | 1 | 1 | 2 | |
| NS SU 346 Ab | | | | | | | | | 1 | 1 | |
| NS SU 347 Ab | | | 1 | 60 | | | | | | 61 | |
| NS SU 351 Sb | 1 | | | | | | | | 1 | 2 | 1P |
| NS SU 356 | | | | | | | | | | 0 | 2P |
| **TOTAL starch granules per morphotype** | 24 | 29 | 102 | 114 | 3 | 40 | 2 | 13 | 63 | 390 | 36 |

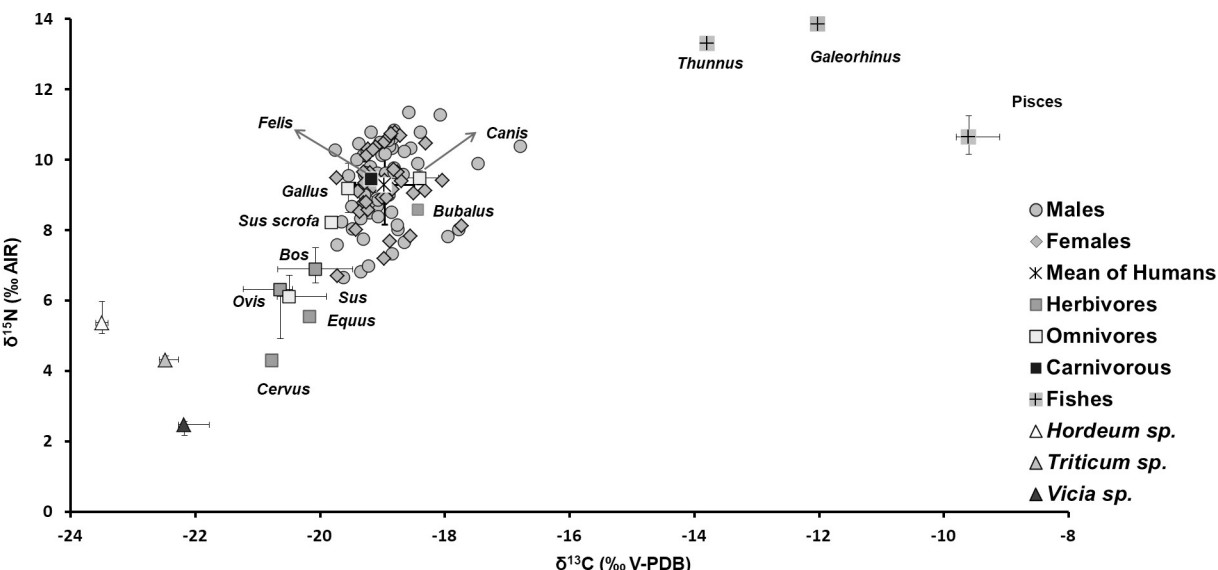

**Fig 1. Human, faunal and seed isotope analysis.** Bivariate plot of the Santa Severa human, faunal and seed stable isotope data; where multiple individuals of a single faunal or plant species were measured, the median and 25th and 75th percentile levels are shown.

The protein samples extracted from human and faunal bones exhibited C:N atomic ratios between 3.1 and 3.3 and between 3.1 and 3.6, respectively, and sufficient yield values (for humans, ranging from 2 to 43%; for animals, ranging from 2 to 38%). Twelve humans and seven animals were discarded for this study (specimens highlighted in red in Table 1), because of the lack of consistency between replicate samples (standard deviation of over 0.2‰). The isotopic results for the eight plant seeds from the site are indicated in Table 2.

The isotopic ratios obtained for the faunal samples exhibited high variability even for specimens attributable to the same species (Fig 1).

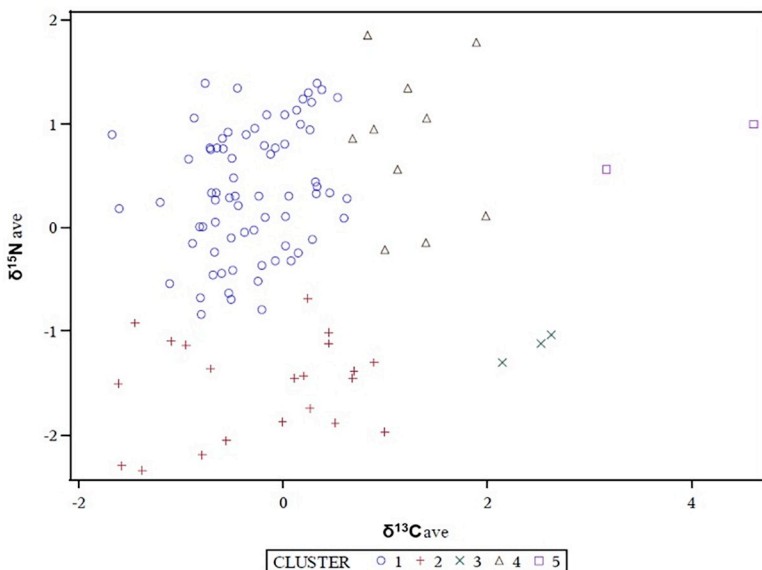

**Fig 2. Human isotope analysis.** Bivariate plot of average linkage cluster analysis for the human stable isotope data from Santa Severa with five groups identified.

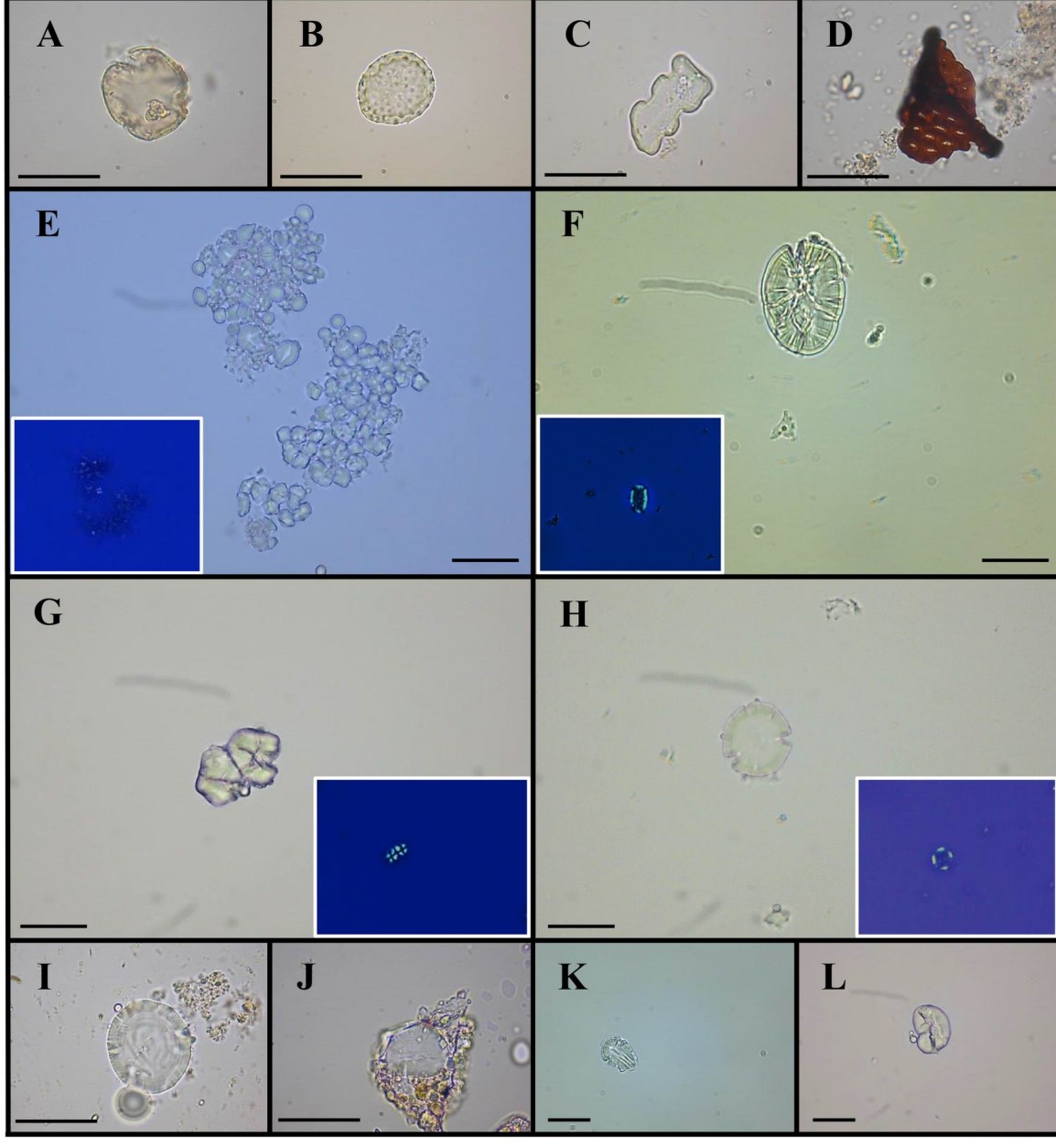

**Fig 3. Plant microremains and starch granules at LM.** Representative images of microdebris found in dental calculus samples: Oleaceae pollen grain (A); Chenopodiaceae pollen grain (B); Poaceae phytolith (C); fragment of Asteraceae inflorescence perforation plate (D); aggregate of *Avena* sp. starch granules and relative polarized image (E); Fabaceae starch granule and relative polarized image (F); *Sorghum* sp. starch granules and relative polarized image (G); Triticeae starch granule and relative polarized image (H); Triticeae starch granules (I and J); Fagaceae starch granule (K); Fabaceae starch granule (L). The scale bar indicates 30 μm.

The specimens placed at the extremes of the plot, deer (*Cervus* sp.) and unidentified fishes (Pisces), showing the lowest and the highest $\delta^{13}$C and $\delta^{15}$N values among all samples, were considered as examples of herbivorous terrestrial and marine animals, respectively. For this reason, they were used as reference values for the interpretation of the remaining animal and human samples. Overall, herbivores demonstrated $\delta^{13}$C and $\delta^{15}$N mean values of -20.2‰ (S.D. 1.3‰) and 6.3‰ (S.D. 1.3‰), respectively. These values are characteristic of animals feeding

on $C_3$ plants. The cattle (*Bos* sp.), sheep (*Ovis* sp.), horse (*Equus* sp.), and buffalo (*Bubalus* sp.) all possess $\delta^{13}C$ and $\delta^{15}N$ values enriched with respect to those observed for the deer (*Cervus* sp.). A single cat represents the only obligate carnivore: its isotopic values ($\delta^{13}C$ = -19.2‰, S. D. 0.1‰, $\delta^{15}N$ = 9.5‰, S.D. 0.2‰) are enriched by about +1‰ in $\delta^{13}C$ and +3‰ in $\delta^{15}N$ compared to herbivores, and are coherent with its relative trophic position (Fig 1). *Sus* sp. samples, expected to possessed isotope values supporting an omnivorous diet, showed measured ratios (mean $\delta^{13}C$ = -20.4‰, S.D. 0.5‰, $\delta^{15}N$ = 6.4‰, S.D. 1.1‰) very close to those observed for sheep, cattle and horse, suggesting they had more likely a herbivorous diet, with the exception of the wild boar that showed values compatible with those observed for omnivores (Fig 1). As anticipated, four chickens (*Gallus* sp.), and two dogs (*Canis* sp.) showed isotope values typical of omnivores, and the marine fishes (3 unidentified samples, 2 specimens of the Sparidae family, 1 *Labrus* sp. and 1 *Sparus* sp.) were more enriched in $^{13}C$ compared to terrestrial species, as to be expected ($\delta^{13}C$ = -9.3‰, S.D. 1‰; $\delta^{15}N$ = 10.7‰, S.D. 0.7‰). *Galeorhinus* sp., a shark genus widely distributed in subtropical areas, represented the consumer at the upper trophic level ($\delta^{13}C$ = -12.0‰; $\delta^{15}N$ = 13.9‰).

Finally, the charred seeds revealed isotope values compatible with $C_3$ photosynthesis and with a basal trophic level. The $\delta^{15}N$ value of *Vicia* (mean $\delta^{15}N$ = 2.4‰, S.D. 0.4 ‰), a legume, was low, as expected, since these plants are not dependent on soil for nitrogen uptake [68–70].

For humans (n = 108), $\delta^{13}C$ values ranged between -19.8‰ to -16.8‰ (mean -19.0, S.D. 0.5‰), while $\delta^{15}N$ values ranged from 6.7‰ to 11.4‰ (mean 9.3, S.D. 1.1‰), as reported in Fig 1. In order to evaluate possible dietary variations between sexes and across 5 selected age groups (0–17 years seven individuals; 18–30 years 33 individuals; 31–40 years 22 individuals; 41–50 years 12 individuals; >50 years 4 individuals) isotope data were statistically compared by Wilcoxon and the Kruskal-Wallis tests. No significant differences were found between the isotopic values for males (n = 62) and females (n = 37), for $\delta^{13}C$ (Wilcoxon test: *p*-value = 0.6522) or $\delta^{15}N$ (Wilcoxon test: *p*-value = 0.8203) ratios. Similar results were obtained on the different age groups (for $\delta^{13}C$ Kruskal-Wallis test: *p*-value = 0.4746, for $\delta^{15}N$ Kruskal-Wallis test: *p*-value = 0.2056). To differentiate between individuals with a similar diet, average linkage cluster analysis was performed on the basis of isotope data. A total of five different groups of samples were identified (Fig 2).

The first group (cluster 1) included 71 individuals; in this context, $\delta^{13}C$ values ranged from -18.7‰ to -19.8‰ (mean = -19.1‰, S.D. 0.2‰) and $\delta^{15}N$ values from 8.3‰ to 10.8‰ (mean = 9.7‰ S.D. 0.7‰). Nitrogen isotope values measured for human specimens were enriched by at least 2.0‰ compared to herbivores, showing a mean human-herbivore difference ($\Delta^{15}N$) of 3.4‰. The second group (cluster 2) was made up of 21 individuals; for these samples, $\delta^{13}C$ values ranged between -18.5‰ and -19.7‰ (mean = -19.1‰, S.D. 0.4‰) and $\delta^{15}N$ values were between 6.7‰ and 8.9‰ (mean = 7.6‰, S.D. 0.6‰). The mean human-herbivore difference ($\Delta^{15}N$) was 1.3‰. The third set (cluster 3) comprised 3 individuals with high $\delta^{13}C$ values; $\delta^{13}C$ ranged between -17.9‰ to -17.7‰ (mean = -17.8‰, S.D. 0.1‰) and $\delta^{15}N$ from 7.8‰ to 8.1‰ (mean = 8.0‰, S.D. 0.2‰). The mean human-herbivore difference ($\Delta^{15}N$) was 1.7‰. The forth cluster (cluster 4) counts 10 individuals, $\delta^{13}C$ ranged from -18.6‰ to -18.0‰ (mean = -18.4‰, S.D. 0.2‰) and $\delta^{15}N$ from 9.0‰ to 11.4‰ (mean = 10.2‰ S.D. 0.8‰) with a mean human-herbivore difference ($\Delta^{15}N$) of 3.9‰. The last set (cluster 5) includes 2 individuals with the highest $\delta^{13}C$ values, where $\delta^{13}C$ ranged between -16.8‰ to -17.5‰ (mean -17.2‰, S.D. 0.5‰) and $\delta^{15}N$ from 9.9‰ to 10.4‰ (mean = 10.2‰ S.D. 0.4‰) with a mean human-herbivore difference ($\Delta^{15}N$) of 3.9‰.

To explore potential differences in diet linked with burial practice, individuals buried in earthen graves were statistically compared to those interred in sarcophagi and no differences were found for either $\delta^{13}C$ or $\delta^{15}N$ (Wilcoxon, $\delta^{13}C$: *p*-value = 0.5312, $\delta^{15}N$: *p*-value = 0.7830).

## Dental calculus analysis

Dental calculus was collected from 94 individuals. Ancient DNA was extracted from the dental calculus of 52 individuals and specific barcode genes were amplified and sequenced, in order to investigate the consumption of specific animal food sources: sheep, chicken, cattle, pig and fish. The spike PCR control showed that no sample contained PCR inhibitors. For each individual, positive amplicons were reported in Table 3.

Overall, animal aDNA was detected in a large number of samples, sheep was present in 55.8% of the analysed individuals, cattle in 34.6% and pig in 25%. Chicken genes were, however, detected only in 11.5% of the analysed sample. Marine fish aDNA was detected in 88.5% of the specimens. These results did not show differences related to age, sex or social status of the individuals inferred from the type of burial.

LM analysis of 94 individuals indicated the presence of plant micro-remains in 70 samples (74.5%). This investigation revealed the presence of several types of micro-debris, as reported in Table 4.

In total, 22 Poaceae phytoliths, 3 calcium oxalate crystals, 2 fragments of Asteraceae inflorescence, 8 pollen grains (3 Oleaceae, 1 Chenopodiaceae, 1 Urticaceae, 3 not determined) and 390 starch granules were observed in the whole population. Only 327 starches could be taxonomically identified and assigned to eight morphotypes, on the basis of morphological and morphometric criteria using both our experimental collection [62] and the International Code for Starch Nomenclature [71]. Each morphotype is described below.

*Morphotype I* (tot. 24 micro-remains) showed the typical features of Fabaceae granules. They were reniform, oval to elongated in shape and ranged in size between 7–35 μm in length and 3–30 μm in width. *Hilum* was not detectable (obscured). Clear concentric *lamellae* were present and, in some granules, a longitudinal crack in the amorphous central area could be observed. Some grains appeared ascribable to starches of *Pisum* L. and *Vicia* L. genera.

*Morphotype II* (tot. 29) consisted of starch granules elongated, drop or oblong-shaped (dimensional range: 5–28 μm in length, 4–20 μm in width). They presented an invisible and eccentric *hilum*, faintly visible *lamellae* and longitudinal fractures, which usually occur in several species of Fagaceae, such as *Quercus ilex* L.

*Morphotype III* (tot. 102) was characterised by a bimodal distribution of disc-shaped grains, typical of grasses of the Triticeae tribe. Among large diagnostic granules (size range: 6–35 μm in length and 4–33 μm in width), 18 were reminiscent of *Triticum* sp. L. (e.g. *T. durum* Desf., *T. aestivum* L., *T. dicoccon* Schrank ex Schübl.), 13 of *Hordeum* sp. L. (e.g. *H. vulgare* L.) and 3 of *Secale* sp. L.

*Morphotype IV* (tot. 114) consisted of polyhedral-shape units from one side and dome shaped on the other. Each unit showed a size ranging from 3–11 μm both in length and in width, a centric-indistinct *hilum* and indistinct *lamellae*. Sometimes, they were found in aggregates. These features were attributable to *Avena* sp. L.

*Morphotype V* (tot. 3) revealed a morphology which completely fitted with those of *Daucus carota* L. grains. In detail, spherical shape, size range from 5–10 μm in length to 4–8 μm in width, peculiar multiple radial fractures, slightly visible concentric *lamellae* and centric *hilum* were detected in these granules.

*Morphotype VI* (tot. 40) was characterized by polyhedral-shape units with blunted edges (size range: 2.5–5 μm both in length and in width). This distinctive trait, associated to a centric-distinct *hilum* and indistinct *lamellae*, suggested that these starches belonged to *Panicum* sp. L.

*Morphotype VII* (tot. 2) showed polyhedral units (size range from 1 to 2.6 μm both in length and in width) with concave faces, acute edges and typical bright boundaries. *Hilum* was centric, while *lamellae* appeared indistinct. All these features indicated *Piper* sp. L. starches.

*Morphotype VIII* (tot. 13) consisted of ovoidal granules with some peculiar flattened surfaces. Their size ranged from 5–18 μm in length to 4–14 μm in width and radial fissures starting from the centric *hilum*. These characteristics were consistent with *Sorghum* sp. Moench starches.

Morphotypes III and IV (Poaceae starches) were identified in most of the individuals, followed by morphotypes I and II (Fabaceae and Fagaceae starches). Particular attention was paid to individual NS SU 321 whose tartar contained 21 starch granules mainly ascribable to morphotype I and II. No difference was evidenced among individuals of different age at death and sex. Representative images of microfossils found in tartar samples are reported in Fig 3.

GC-MS analysis was performed only on the samples who presented sufficient dental calculus (68 individuals). In S5 Table, the molecules identified by this approach are listed and clustered in biochemical classes per individual. Food categories or specific plants were deduced by associating the recognized compounds. All chromatographic profiles revealed a significant presence of *n*-alkanes and *n*-alkenes ($C_6$-$C_{35}$). Generally, monounsaturated (e.g. docosenoic and octadecenoic acids) and polyunsaturated (e.g. octadecadienoic and octadecatrienoic acids) fatty acids, ω3-fats (e.g. eicosapentaenoic and docosahexaenoic acids) and lactose were the most recurrent molecules detected in the population, followed by cholesterol and secondary metabolites indicating aromatic herbs (e.g. anisole, alpha-cubebene, estragole, santolina triene, beta-copaene, dehydroelsholtzia ketone). The identification of phytosterols (e.g. stigmasterol, sitosterol, campesterol), vitamins (e.g ascorbic acid and tocopherol) and other classes of plant markers (e.g. lactones, glucosinolates, phenolics) suggested the key role of vegetables and fruits in the diet of the population. Moreover, the identification of tartaric and gallic acids, pyrocathecol, and pyrogallol hypothesized a possible consumption of wine. Evidence of alkaloids (e.g., pseudopelletierine, ephedrine) and terpenic compounds (e.g. bisabolene, scoparone, alpha-bisabolol, dihydroartemisinin, all ascribable to Asteraceae) supported the use of aromatic and medicinal plants. Finally, detection of ergosine and bovinocidin in two samples suggested a fungal contamination of stored foods. No difference was evidenced among individuals showing different age at death, social class and sex.

## Discussion

Here we have applied a suite of scientific techniques to the skeletal remains of the Medieval community of Santa Severa (7th-15th century CE) in order to highlight their culture and food habits. This research integrates the available morphological data [24] with the results obtained by the present original and innovative multidisciplinary approach, with the aim to reconstruct the osteobiography of one of the largest Italian Medieval burial populations.

Faunal specimens showed high variability in their isotopic values. The differences observed between the deer (*Cervus* sp.) specimen and the other herbivores (6 cattle *Bos* sp., 5 sheep *Ovis* sp., 1 horse *Equus* sp., and 1 buffalo *Bubalus* sp.) could be attributable to the different environments in which these animals lived [31]. Deer tend to live in wooded environments, while domestic animals would have lived in closer proximity to humans and would have been managed by them. Therefore, the observed 15N enrichment in sheep and cattle could be ascribable to both human activities and potentially proximity to the sea [31]. Some crop management practices (e.g. manuring) could increase δ15N values in soil and plants, while plants might also have absorbed nitrogen of marine origin, due to the "sea spray effect" [72–73]. *Sus* sp. samples had dietary signals more similar to those observed for herbivores than omnivores, as in earlier studies [74] although the wild boar (*Sus scrofa*) had significantly different values and presented an omnivorous diet. In Medieval North-Western Europe, the diet of pigs was mainly based on terrestrial plants and human refuse [17, 18, 75], although in some areas, including Latium, pigs

were free to roam in the uncultivated land surrounding the city [17] where they could consume acorns [15, 17].

The stable isotope data for humans indicated an omnivorous diet with a contribution of both animal products and plants. The high variability observed in nitrogen isotopic values (ranging from 6.7‰ to 11.4‰) seemed to witness a probable differential access to food sources within the population.

The $\delta^{13}C$ values for humans were compatible with a diet mainly based on consumption of $C_3$ plants; however, the highest $\delta^{13}C$ values ($\delta^{13}C > 18$‰) observed in some specimens (cluster 3) could also suggest a contribution of $C_4$ plants to the diet. This is not surprising as during the Middle Ages, $C_4$ crops such as sorghum and millets, were grown and consumed due to their ease of cultivation and relatively high yields [15, 76–77]. These results are in line with the archaeobotanical results from dental calculus described below.

Given the coastal location and late Medieval date, it is surprising that the isotope data do not reflect a significant contribution of marine protein to the human diet. The identification of a marine isotopic signal with typical enrichment in both $^{13}C$ and $^{15}N$ however, is problematic in the Mediterranean context where the contribution of $C_4$ terrestrial protein exists, as Mediterranean fish tend to have lower $\delta^{13}C$ and $\delta^{15}N$ values than those of the Atlantic [78–80]. Fish consumption can not be totally excluded as marine fish aDNA and ω3-fatty acids were detected in dental calculus, although the latter are also widely abundant in plants. Aside from the issue of $C_4$/marine diets, exhibiting similar values [78–80], it may also be possible that the consumption of marine protein sources was not sufficient to induce a significant shift of isotopic values [81].

Only three individuals (NS SU 124 Aa, PR SU 262 Aa and NS SU 321) possessed isotopic values potentially compatible with fish consumption ($\delta^{13}C$ -16.8‰, -17.5‰, -18.1‰ and $\delta^{15}N$ 10.4‰, 9.9‰, 11.3‰ respectively). However, due to the presence of non-specific stress markers (e.g. cribra), as well as infections (e.g periostitis and osteomyelitis) in two of these specimens (NS SU 124 Aa, NS SU 321) [24], the hypothesis of higher nitrogen values for these individuals being affected by potential nutritional stress cannot be totally excluded [82]. It has been documented that starvation may stimulate gluconeogenesis, and therefore the production of glucose from non-carbohydrate sources [47, 83–87], causing $^{15}N$ enrichment in body tissues [83, 88]. The isotope values for NS SU 321 in particular, however, may be reflective of a high *status* diet. This individual, atypically buried in a sarcophagus with a cross and a stone cushion [23], presented the markers for all the tested food sources (bovine, swine, ovine, chicken and fish: Table 3), according to genetic analysis of dental calculus. The consumption of high trophic level protein (meat and to some extent fish) was affiliated with a high-status identity in Medieval society [15]. Although no statistical difference was detected between sex and age at death groupings, a probable internal division of the society was suggested by the average linkage cluster analysis (Fig 2). The clusters differ on both carbon and nitrogen isotopic signatures. In particular, as regard the former, three of them (cluster 1, cluster 2, and cluster 4) include individuals possessing isotopic values compatible with a prevalent consumption of $C_3$ plants whereas carbon isotopic values of the other two clusters (cluster 3 and cluster 5) are compatible with the intake of $C_4$ species. Similarities in $\delta^{13}C$ values, however do not square with nitrogen isotopic ratios. The lowest nitrogen detected in clusters 2 and 3, both also therefore characterized by the lowest human-herbivore $^{15}N$ enrichment (1.3‰ and 1.7‰, respectively) suggesting a more vegetarian diet with a reduced animal protein intake.

Alternatively, a higher contribution of animal (mainly terrestrial) protein sources in diet is indicated by the higher $\delta^{15}N$ values observed for the individuals grouped into the other three clusters that also posess higher human-herbivore differences (3.4‰, 3.9‰, and 3.9‰ respectively). None of the clusters are defined by individuals showing clear isotopic signatures of fish

consumption. The above mentioned three individuals for which fish consumption is most likely are grouped in clusters 4 (NS SU 321) and 5 (NS SU 124 Aa, PR SU 262 Aa). These isotopic clusters, however, are not correlated with the different typologies of burials identified in the archaeological site (e.g. simple earthen graves and sarcophagi made of Etruscan and Roman re-used tuff stones), as they are equally represented within the clusters. The variability of nitrogen isotopic values detected in the population may instead be the result of the wide time span that these burials represent (about 900 years, 7[th]-15[th] century CE). During the Early Middle Ages (6[th]-10[th] century), the diet, in Italy, was typically associated with the consumption of a large amount of animal protein (meat and fish), in association with plant foods [77, 89]. During the Late Medieval period (11[th]-15[th] century), the intake of appreciable amounts of animal protein became a privilege of the upper classes [15, 77]. A potential dietary transition between the Early and Late Middle Ages may also be the result of ecclesiastic law requiring abstinence from meat promoting, instead the consumption of fish during the later period [2, 7–18]. Unfortunately, we are unable to further explore the chronological element, as it has not been possible to identify which burials derived from the early and later Medieval period.

The isotope data for published Medieval populations from Italy is generally very variable (S6 Table). The highest $\delta^{13}$C values can be observed by North-Eastern sites, dating to the early Medieval period (6[th]-11[th] centuries), like Romans d'Isonzo, Cividale Gallo, Cividale Santo Stefano and Mainizza [90], as well as Cosa (11[th]-13[th] centuries; [86]) from Central Italy. The observed enrichment in $^{13}$C of these samples, with respect to Santa Severa, is attributable to a higher consumption of $C_4$ plants (e.g. millet) [86, 90]. Indeed, North-Eastern Italy, experienced an onward decline in bread consumption which was substituted by soups [90]. With regard to nitrogen isotopes, the population from Santa Severa was more enriched in $^{15}$N than several sites across Medieval Italy, namely Romans d'Isonzo and Mainizza in the North East [90], Colonna (8[th]-10[th] centuries; [19]) in Central Italy, and Montella (13[th]-15[th] centuries; [91]) in Southern Italy. The North-Eastern Medieval populations were characterized by a low animal protein intake, (although freshwater fish intake has been suggested for some [90]), whereas at Montella (Southern Italy), the analysed population was associated with a Franciscan friary and so there is the potential that this monastic population followed a distinctive dietary regime [91]. Although Colonna and Santa Severa are present in the same region of Italy, a differential access to nutritional sources related to sex and age at death in this case, was detected at Colonna [19]. In particular, at Colonna, adult male individuals demonstrated a greater intake of animal protein with respect to females and juveniles, who followed a more vegetarian diet [19].

When considering the results from dental calculus, it should be borne in mind that the number of starch granules and molecules present in dental calculus of an individual is not necessarily proportional to the amount of foods consumed in life. However, these data, considered at population level, represent direct evidence of the substances that entered the oral cavity [92–93]. LM analysis indicated the presence of $C_3$ cereals (e.g. *Avena* sp. and Triticeae), Fabaceae (e.g. *Vicia* sp., *Pisum* sp.) and Fagaceae (e.g. *Quercus* sp. L.) seeds, in more than half of the individuals. Moreover, for some specimens, we hypothesized the intake of $C_4$ caryopses (e.g. *Sorghum* sp. and *Panicum* sp.). In general, the finding of these types of starches is in keeping with the carpological remains discovered in an oven of the same archaeological site [23], supporting the use of these plant species as the main sources of carbohydrates. The presence of Fagaceae starches is not surprising because acorns have been recognised as a nutritional resource containing sugars, vitamins and proteins since the Roman period. After the removal of indigestive tannins, acorns were powdered and used to ennoble cereal flours or prepare astringent and anti-diarrheal decoctions [22, 93–96]. Similarly, legumes represented both an important food that could be preserved for a long time after drying, and a source of nutraceutical compounds

applied in dermatology [97–98]. Moreover, these species were also excellent feed crops, already used in the Roman three-field rotation strategy (together with cereals and fallow), for their ability to increase soil nitrogen [99–100]. The high number of starch grains that could not be identified may be the result of cooking processes, grinding procedures or exposure to Ptialin enzyme activity [101]. The finding of Oleaceae, Chenopodiaceae, Urticaceae pollen grains and fragments of Asteraceae inflorescence testified the existence of these plant families in the studied area. The possible use of these species in dietary and/or ethnopharmacological traditions cannot be excluded. For instance, *Urtica dioica* L. (nettle) and *Parietaria officinalis* L. (upright pellitory) were widely employed for medicinal purposes, as diuretic, emollient and expectorant [102].

The GC-MS analysis on dental calculus revealed a great variety of *n*-alkanes and *n*-alkenes, probably deriving from degradation of both plant and animal food molecules or/and oral microbiota [103–104]. The detection of mono/poly-unsaturated fatty acids, phytosterols and plant vitamins suggested a diet based on plant foods, such as seeds (e.g. cereals, confirmed by LM), vegetables (e.g. Brassicaceae, also supported by the presence of glucosinolates) and fruits (e.g. of Rosaceae family due to the presence of lactones) [3, 105–107]. Moreover, ω3-fatty acids could be associated with ingestion of plant and/or aquatic resources (e.g. dried fruits, seaweeds, molluscs, blue fishes) [108–109]. A great part of the community also had lactose present in their dental calculus, the main sugar of milk and dairy products, confirming the fundamental role of these animal derivatives in the Medieval diet [110–111]. Some individuals presented secondary metabolites of aromatic plants, such as Apiaceae (e.g. dill) and Lamiaceae (e.g. sage), leading us to hypothesize that they employed these herbs as food preservatives and/or taste regulators [112]. Our data confirmed the consumption of wine (*Vitis vinifera* L.) [113–115]. The pseudopelletierine, an alkaloid found in NS SU 124 Aa sample, proved the ingestion of *Punica granatum* L. bark [116]. Indeed, various portions of pomegranate were used in folk medicine, including Egyptian one; flowers for treating diabetes; fruits for expelling parasites; seeds and fruit peels for managing diarrhoea; bark and roots as coagulants and anti-ulcer remedies [116–118]. Pedanius Dioscorides, a Greek physician (40–90 AD), and several Roman documents report the recipes for the preparation of mouthwashes based on pomegranate rind and bark [119–120]. Indeed, it has actually been recently scientifically demonstrated that the extracts of this species prevent the dental plaque development [118, 121]. A typical marker of *Elsholtzia* sp. Willd. (dehydroelsholtzia ketone) and a peculiar chemical compound of *Ephedra* sp. L. (ephedrine) indicated a possible intake of decoctions based on these plants, probably to treat respiratory and gastro-intestinal disorders [122–123].

In three individuals (NS SU 27 Ac, NS SU 115, NS SU 290) we found the artemisinin, a sesquiterpene lactone typically synthesized in *Artemisia annua* L. (Asteraceae family) and presenting antimalarial properties [124]. As Santa Severa Castle was surrounded by marshy areas during the Medieval period [125], a condition which favour the survival of the *Plasmodium falciparum*, our results might sustain the existence of malaria in Central Italy and its therapeutic treatment by the use of this plant. Unfortunately, no distinctive signs on bone remains can document and support the previous hypothesis. As extensively investigated by earlier studies [126–127] malaria determines a haemolytic anaemia that may result in several modifications on human bones even though it can be considered one of the possible causes of *cribra orbitalia* [126–129]. Indeed, *cribra orbitalia*, represents one of the most frequently recorded alterations in archaeological skeletal collections; it pertains to the presence of small foramina in the orbit vaults, due to the expansion of the diploe accompanied by the narrowing of the outer bone cortex as a consequence of bone marrow hypertrophy [127, 130–132]. *Cribra orbitalia* has been traditionally considered a clue of iron deficiency anaemia [127, 130–132], it is worth noting, however, that defining the exact cause of anaemic conditions could be challenging as they

could have been determined by several factors, such as iron or ascorbic acid depletion, other nutritional deficiencies, infections, or even by a combination of these elements [47, 127, 133–136]. Moreover, a similar bone morphology could be also determined by scurvy and chronic infections [47, 127, 130–132]. As reported by Gowland and Western [127] malaria can be directly diagnosed only by the detection of Plasmodium species DNA in human skeletal remains [127], as successfully carried out in Egyptian mummies [127, 137–138] and in individuals recovered in an infant cemetery in Umbria (Italy, 5th century CE; [139]) [127]. The use of Asteraceae, as food or medicinal sources [140], in these individuals was also confirmed by the detection of bisabolol (in NS SU 27 Ac), bisabolene (in NS SU 27 Ac and NS SU 115) and the LM survey of fragments of capitulum inflorescence (in NS SU 115 and NS SU 290).

Lastly, the presence of two molecules, the bovinocidin (produced by food-contaminating molds such as *Aspergillus* sp. P. Micheli ex Haller) [141] and the ergosine (an alkaloid synthesized by *Claviceps purpurea* (Fr.) Tul., the fungal parasite of Poaceae) [142], could indicate the use of inadequate practices in food conservation and the diffusion of Graminaceae phytopathogens in that historical context.

## Conclusions

In conclusion, the cutting-edge multidisciplinary approach applied in this work has enabled a detailed reconstruction of the dietary habits of one of the largest Italian Medieval populations analysed to date. The isotope analyses performed on human, faunal and seed samples, combined with archeobotanical and molecular analysis of dental calculus, revealed that the population generally adopted an omnivorous diet, in which plants and fruits played a prominent role. With regard to animal protein intake, cattle, sheep and pig meat were consumed with a minor contribution of chicken. Consumption of $C_4$ plants (millet and/or sorghum) was also evidenced in a few individuals. Significantly, the application of multiple complementary methodologies revealed a potential intake of marine foodstuffs as dietary resource for this Medieval population. The consumption of marine foods was not clearly detectable using stable isotope evidence, highlighting the limitations of using this method alone in the Mediterranean context. Archaeobotanical analysis further revealed the presence of artemisinin that may be an indication of the presence of malaria in the area of Santa Severa. The dietary (e.g. cereals, Brassicaceae) and medicinal (e.g. pomegranate, *Ephedra* sp.) uses of other plant species was also hypothesised. This innovative archaeobotanical approach combined with the anthropological data may provide interesting information about presence and diffusion of pathologies and diseases in the past, beyond those are not directly detectable on skeletal remains.

## Supporting information

**S1 Table. Biological profile of the Medieval individuals subjected to dental calculus analyses.** Each sample was associated to a code, reporting NS (for identifying the archaeological area of "Casa del Nostromo") and a number (relative to the stratigraphic unit, SU), according to [24]. For each specimen, estimated sex (F, female; M, male; ND, sex determination was not determinable for the lack of diagnostic elements due to their bad conservation; IND, subadults with not determined sex because of sexual immaturity), estimated age at death (GA, generic adult individual) and district of the masticatory apparatus used for dental calculus sampling were reported. In particular, sampled teeth and relative surfaces were codified according to the Universal Teeth Numbering System [143] (B, buccal; L, lingual; M, mesial; D, distal). Finally, sample weight was indicated in grams and the analytical techniques applied on each sample were marked with an X.
(DOCX)

**S2 Table. Light microscopy results of the laboratory contamination tests.**
(DOCX)

**S3 Table. Light microscopy results obtained from the washing water applied on ancient dental calculus before the cleaning procedure.**
(DOCX)

**S4 Table. The primers used for Spike PCR (L15996 and H16401) and detection of aDNA relative to bovine (Bov84/90-F/R), pig (Sus85-F/R; Sus98-F/R), ovine (Ovis-F/R), chicken (Gall-F/R) and fishes (Fish_miniA_F/R; Fish_miniC_F/R) were shown.** In addition, length of PCR amplicons and annealing temperatures were also reported.
(DOCX)

**S5 Table. The chemical compounds detected in dental calculus by GC-MS analysis were listed and clustered in biochemical classes for each sample.**
(DOCX)

**S6 Table. Mean of $\delta^{13}$C and $\delta^{15}$N of human samples recovered in Santa Severa archaeological site and in other coeval Italian sites from literature.** For each sample. chronology, sample size and relative reference were reported [7, 8, 19, 20, 21, 86, 90, 91, 144–145].
(DOCX)

## Acknowledgments

The authors thank Harry Robson (University of York) for his help in identifying the fish remains and we are grateful to Luke Spindler (University of York) for his help in collagen extraction for fish.

The authors sincerely thank the editor and the anonymous reviewer for the valuable comments that contributed to improve the present paper.

## Author Contributions

**Conceptualization:** Angelo Gismondi, Antonella Canini, Olga Rickards, Cristina Martínez-Labarga.

**Data curation:** Angelo Gismondi, Marica Baldoni, Micaela Gnes, Gabriele Scorrano, Alessia D'Agostino, Gundula Müldner, Alessandra Nardi, Flavio Enei, Michelle Alexander, Cristina Martínez-Labarga.

**Formal analysis:** Angelo Gismondi, Marica Baldoni, Micaela Gnes, Gabriele Scorrano, Alessia D'Agostino, Gabriele Di Marco, Giulietta Calabria, Michela Petrucci, Gundula Müldner, Matthew Von Tersch, Alessandra Nardi, Cristina Martínez-Labarga.

**Funding acquisition:** Flavio Enei.

**Investigation:** Angelo Gismondi, Marica Baldoni, Micaela Gnes, Gabriele Scorrano, Alessia D'Agostino, Cristina Martínez-Labarga.

**Methodology:** Angelo Gismondi, Marica Baldoni, Micaela Gnes, Gabriele Scorrano, Alessia D'Agostino, Gabriele Di Marco, Gundula Müldner, Michelle Alexander, Cristina Martínez-Labarga.

**Project administration:** Cristina Martínez-Labarga.

**Resources:** Antonella Canini, Olga Rickards.

**Supervision:** Cristina Martínez-Labarga.

**Validation:** Cristina Martínez-Labarga.

**Visualization:** Cristina Martínez-Labarga.

**Writing – original draft:** Angelo Gismondi, Marica Baldoni, Micaela Gnes, Cristina Martínez-Labarga.

**Writing – review & editing:** Angelo Gismondi, Marica Baldoni, Micaela Gnes, Gabriele Scorrano, Alessia D'Agostino, Gabriele Di Marco, Giulietta Calabria, Michela Petrucci, Gundula Müldner, Matthew Von Tersch, Alessandra Nardi, Flavio Enei, Antonella Canini, Olga Rickards, Michelle Alexander, Cristina Martínez-Labarga.

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
