## [Decision Letter · Decision Letter 0]

2 Oct 2019

PONE-D-19-21430

A multidisciplinary approach for investigating dietary and medicinal habits of the Medieval population of Santa Severa (7th-15th centuries, Rome, Italy)

PLOS ONE

Dear Prof. Martínez-Labarga,

Thank you for submitting your manuscript to PLOS ONE. After careful consideration, we feel that it has merit but does not fully meet PLOS ONE’s publication criteria as it currently stands. Therefore, we invite you to submit a revised version of the manuscript that addresses the points raised during the review process.

We would appreciate receiving your revised manuscript by Nov 16 2019 11:59PM. To enhance the reproducibility of your results, we recommend that if applicable you deposit your laboratory protocols in protocols.io, where a protocol can be assigned its own identifier (DOI) such that it can be cited independently in the future. For instructions see: http://journals.plos.org/plosone/s/submission-guidelines#loc-laboratory-protocols

We look forward to receiving your revised manuscript.

Kind regards,

David Caramelli, Ph.D

Academic Editor

PLOS ONE

Journal Requirements:

1. In your manuscript, please provide additional information regarding the specimens used in your study. Ensure that you have reported specimen numbers and complete repository information, including museum name and geographic location.

For more information on PLOS ONE's requirements for paleontology and archaeology research, see https://journals.plos.org/plosone/s/submission-guidelines#loc-paleontology-and-archaeology-research.

Reviewers' comments:

Reviewer's Responses to Questions

**Comments to the Author**

1. Is the manuscript technically sound, and do the data support the conclusions?

Reviewer #1: Partly

2. Has the statistical analysis been performed appropriately and rigorously? 

Reviewer #1: Yes

3. Have the authors made all data underlying the findings in their manuscript fully available?

Reviewer #1: Yes

4. Is the manuscript presented in an intelligible fashion and written in standard English?

Reviewer #1: Yes

5. Review Comments to the Author

Reviewer #1: With your research, Gismondi et al. reconstructed dietary and medicinal habits of a Medieval Italian population using a multidisciplinary approach that involves stable isotope analysis from bone proteins, DNA and plant microremains analyses as well as gas chromatography coupled with mass spectrometry on dental calculus. The results presented in the paper are useful to reconstruct the lifestyle and the health status of a medieval population from Central Italy, and represent a further confirmation of the amount of multidisciplinary information that dental calculus can potentially provide.

I have few comments that the authors should take into account to improve the study.

A major issue that I found is related to the molecular and archaeobotanical analyses on dental calculus:

- Sample collection and decontamination. I suggest adding a supplementary table with archaeological information (for instance sex and estimated age at death) of each individual analyzed and information about calculus sampling (i.e tooth and weight of the calculus).

- Page 6 line 147: you assert that after the cleaning procedures, dental calculus was examined by optic microscopy “to confirm the efficacy of these decontamination methods”. What are the parameters taken into account to establish the efficacy of the decontamination methods used?

- In the text it is not clear if the molecular, archaeobotanical and gas-chromatography analyses were carried out on the same individual; I believe that you did, since dental calculus was collected from 94 individuals, aDNA was extracted from 52 individuals, LM analysis was conducted on all the specimens and GC-MS was performed on 68 individuals. However, I suggest authors adding a brief explanation in the main text and a supplementary table with the analyses performed on each sample.

For samples subjected to multiple types of analyses, how did you split the calculus? Ideally, dental calculus samples from the same teeth should be used for the same analysis, because of variation in DNA content and residues. Alternatively, the calculus sampled from different teeth may be pooled and subsequently divided into sub-samples. Specify in the text.

- Results section.

Stable isotope analysis. Pages 16-17 lines 275-294. An average linkage cluster analysis was performed to differentiate between individuals with the similar diet, and 5 groups were identified. Authors provided description of the isotopic values for each group, but it is not clear what these data mean. I suggest authors a brief explanation/comment in this section or in Discussion section.

Dental calculus analysis. The aDNA analysis were performed using a “traditional approach” PCR-based, and lacks of evidence that the results obtained are authentic ancient DNA data (i.e damage patterns, fragment lengths). On the other hand, the data obtained were confirmed by isotopic and GC-MS analyses except for marine fish. My doubt concerns the authenticity of the PCR results for fishes: indeed, 46 samples out of 52 show positive PCR results and only two (DNA was not investigate on PR SU 262 individual) of these exhibits isotopic values indicative of fish consumption. Taking into account the limitations of using isotopic method in the Mediterranean context, authors have explained this data as “it may also be possible that the consumption of marine protein was not sufficient to induce a significant shift isotopic values”. Is it possible that such low consumption of fish allows to recover fish residues entombed in calcified plaque? Could these PCR results derive from contamination?

Regarding the microremains analysis, authors have identified morphotypes ascribable to plants commonly distributed in the territory. For this reason, it is important verify the occurrence of a possible contamination in analyzed ancient dental calculus, for instance by analysis of a control, sampled from a different part of the skull than teeth.

Regarding the GC-MS analyses. This is not my field of research and I have only one question: are there any authentication criteria to validate the results obtained?

6. PLOS authors have the option to publish the peer review history of their article (what does this mean?). If published, this will include your full peer review and any attached files.

Reviewer #1: No

---

## [Author Response · Author response to Decision Letter 0]

29 Nov 2019

We thank the Reviewer for their insightful and constructive comments. We have carefully considered all comments and have revised and improved our manuscript accordingly, as described below. All the changes are in red in the main text of the re-submitted manuscript copy with track changes.

Comments to the Authors

Reviewer #1: With your research, Gismondi et al. reconstructed dietary and medicinal habits of a Medieval Italian population using a multidisciplinary approach that involves stable isotope analysis from bone proteins, DNA and plant microremains analyses as well as gas chromatography coupled with mass spectrometry on dental calculus. The results presented in the paper are useful to reconstruct the lifestyle and the health status of a medieval population from Central Italy, and represent a further confirmation of the amount of multidisciplinary information that dental calculus can potentially provide.

A: We thank the reviewer for the comment.

R1: I have few comments that the authors should take into account to improve the study.

A major issue that I found is related to the molecular and archaeobotanical analyses on dental calculus:

Sample collection and decontamination. I suggest adding a supplementary table with archaeological information (for instance sex and estimated age at death) of each individual analyzed and information about calculus sampling (i.e. tooth and weight of the calculus).

R1: In the text it is not clear if the molecular, archaeobotanical and gas-chromatography analyses were carried out on the same individual; I believe that you did, since dental calculus was collected from 94 individuals, aDNA was extracted from 52 individuals, LM analysis was conducted on all the specimens and GC-MS was performed on 68 individuals. However, I suggest authors adding a brief explanation in the main text and a supplementary table with the analyses performed on each sample.

A: Thank you for the suggestion. We added a supplementary table (see S1 Table) with all requested information. Moreover, we modified the text as suggested (lines 141-145).

R1: For samples subjected to multiple types of analyses, how did you split the calculus? Ideally, dental calculus samples from the same teeth should be used for the same analysis, because of variation in DNA content and residues. Alternatively, the calculus sampled from different teeth may be pooled and subsequently divided into sub-samples. Specify in the text.

A: We thank the reviewer for the comment. The text was modified as suggested (lines 141-145). Dental calculus was sampled based on the amount available of each individual. Dental calculus collected from different teeth per individual was pooled and subsequently divided into sub-samples to perform all analysis. Indeed, working on several calculus flakes (from the same individual) increases the results that can be extrapolated from various areas of the mouth, where the debris accumulates in a different way and quantity.

R1: Page 6 line 147: you assert that after the cleaning procedures, dental calculus was examined by optic microscopy “to confirm the efficacy of these decontamination methods”. What are the parameters taken into account to establish the efficacy of the decontamination methods used?

A: To investigate if our decontamination protocols were adequate, we chose to observe by light microscopy the washing water of the dental calculus, both before and after the application of these sterilization procedures, as reported in “Sample collection and decontamination” section. The absence of microdebris in the washing water of the samples after the cleaning procedures was considered the proof of the efficacy of the method, considering that the washing water of some samples before the cleaning procedures sometimes showed particles. To satisfy the reviewer we added a specific sentence (lines 157-162) and a table (S3 Table), reporting the results of the contamination tests of the washing water before and after the sterilization procedures. Moreover, we added another table (S2 Table) and sentence in the text (lines 150-151), to provide information about the continuous monitoring of laboratory contamination level. 

R1: Stable isotope analysis. Pages 16-17 lines 275-294. An average linkage cluster analysis was performed to differentiate between individuals with the similar diet, and 5 groups were identified. Authors provided description of the isotopic values for each group, but it is not clear what these data mean. I suggest authors a brief explanation/comment in this section or in Discussion section.

A: We provided a comment on the results of the average linkage cluster analysis in the discussion (lines 463-476).

R1: Dental calculus analysis. The aDNA analysis were performed using a “traditional approach” PCR-based, and lacks of evidence that the results obtained are authentic ancient DNA data (i.e damage patterns, fragment lengths). On the other hand, the data obtained were confirmed by isotopic and GC-MS analyses except for marine fish. My doubt concerns the authenticity of the PCR results for fishes: indeed, 46 samples out of 52 show positive PCR results and only two (DNA was not investigate on PR SU 262 individual) of these exhibits isotopic values indicative of fish consumption. Taking into account the limitations of using isotopic method in the Mediterranean context, authors have explained this data as “it may also be possible that the consumption of marine protein was not sufficient to induce a significant shift isotopic values”. Is it possible that such low consumption of fish allows to recover fish residues entombed in calcified plaque? Could these PCR results derive from contamination?

A: Thanks to the referee for this comment. Concerning the contamination, we believe that our data does not derive from environmental or laboratory contamination. Indeed, we carried out all experiments applying strategies to avoid any contamination; we specified them in Materials and Methods.

To date, there is no evidence about the amount of consumption needed to entrap dietary residues in calcified plaque. The incorporation of foods during biomineral maturation of dental calculus seems to be stochastic and that their preparation seems could affect the dental calculus formation (Radini A., et al., 2017. Beyond food: The multiple pathways for inclusion of materials into ancient dental calculus. Am J Phys Anthropol; 162:71–83.). Therefore, it is possible that low consumption of fish allows to recover fish residues and DNA from calcified plaque. Moreover, we believe that the consumption of fish for the individuals of the present community is supported by other archaeological evidence: i) the geographic location of the area (near the sea); ii) the uncommonly high amount of fish remains, of different species, found in the site (see Table 1).

R1: Regarding the microremains analysis, authors have identified morphotypes ascribable to plants commonly distributed in the territory. For this reason, it is important verify the occurrence of a possible contamination in analyzed ancient dental calculus, for instance by analysis of a control, sampled from a different part of the skull than teeth.

A: Thank you for this comment. We think that it is more correct to observe directly the washing water of the dental calculus (rather than other parts of the skeleton as control) before and after the application of sterilization and decontamination procedures on the sample, as described in the previous answer. Moreover, to check laboratory contamination we always carry out laboratory tests. For these reasons, we added in the text the data obtained from all these experiments (see Table S3 Table and S2 Table). 

R1: Regarding the GC-MS analyses. This is not my field of research and I have only one question: are there any authentication criteria to validate the results obtained?

A: As we have specified in the MS, the mass spectrum of each chemical compound detected from ancient samples was directly compared, by the instrument, with the reference spectra of 192.108 different molecules registered in the NIST LIBRARY 14 (constantly updated). The identified molecules were considered acceptable only if the similarity value of their mass spectra was equal or higher than 85%. It means that the chemical structure of the molecules is well-preserved and authentic. Moreover, the association of these chemical compounds with plant species or food categories is carried out, then, by comparison with literature and scientific food databases (FoodDB, 2013; TGSC, 2015).

---

## [Editor Report · Decision Letter 1]

19 Dec 2019

A multidisciplinary approach for investigating dietary and medicinal habits of the Medieval population of Santa Severa (7th-15th centuries, Rome, Italy)

PONE-D-19-21430R1

Dear Dr. Martínez-Labarga,

We are pleased to inform you that your manuscript has been judged scientifically suitable for publication and will be formally accepted for publication once it complies with all outstanding technical requirements.

With kind regards,

David Caramelli, Ph.D

Academic Editor

PLOS ONE
---

## [Editor Report · Acceptance letter]

20 Dec 2019

PONE-D-19-21430R1 

A multidisciplinary approach for investigating dietary and medicinal habits of the Medieval population of Santa Severa (7th-15th centuries, Rome, Italy) 

Dear Dr. Martínez-Labarga:

I am pleased to inform you that your manuscript has been deemed suitable for publication in PLOS ONE. Congratulations! Your manuscript is now with our production department. 

With kind regards,

on behalf of

Professor David Caramelli 

Academic Editor

PLOS ONE